# $m^6A$ and the NEXT complex direct Xist RNA turnover and X-inactivation dynamics

Guifeng Wei [1] ✉, Heather Coker[1], Lisa Rodermund[1], Mafalda Almeida [1],
Holly L. Roach [1,2], Tatyana B. Nesterova [1] & Neil Brockdorff [1] ✉

X-chromosome inactivation (XCI) in mammals is orchestrated by the noncoding RNA X-inactive-specific transcript (Xist) that, together with specific interacting proteins, functions in *cis* to silence an entire X chromosome. Defined sites on Xist RNA carry the $N^6$-methyladenosine ($m^6A$) modification and perturbation of the $m^6A$ writer complex has been found to abrogate Xist-mediated gene silencing. However, the relative contribution of $m^6A$ and its mechanism of action remain unclear. Here we investigate the role of $m^6A$ in XCI by applying rapid degron-mediated depletion of METTL3, the catalytic subunit of the $m^6A$ writer complex, an approach that minimizes indirect effects because of transcriptome-wide depletion of $m^6A$. We find that acute loss of METTL3 and $m^6A$ accelerates Xist-mediated gene silencing and this correlates with increased levels and stability of Xist transcripts. We show that Xist RNA turnover is mediated by the nuclear exosome targeting complex but is independent of the principal nuclear $m^6A$ reader protein YTHDC1. Our findings demonstrate that the primary function of $m^6A$ on Xist RNA is to promote Xist RNA turnover, which in turn regulates XCI dynamics.

X-chromosome inactivation (XCI) is a developmentally regulated process that evolved in mammals to equalize the levels of X-linked gene expression in XX females relative to XY males[1]. Silencing of a single X chromosome in cells of XX embryos is orchestrated by the X-inactive-specific transcript (Xist), a ~17-kb noncoding RNA, which accumulates in *cis* across the future inactive X (Xi) chromosome[2–5]. Functional elements within Xist RNA have been assigned in large part to tandem repeat blocks labeled A–F that are distributed across the length of the transcript. Most notably, the 5′-end-located A-repeat element has been found to be critical for Xist-mediated gene silencing[6]. Identification of RNA-binding proteins (RBPs) that interact with the A-repeat and other elements has been achieved using both proteomic[7–9] and functional genetic screening[10,11] strategies. These approaches led to the identification of the RBP SPEN as a critical factor for Xist-mediated silencing, functioning by binding to Xist A-repeat region[12] and recruitment of the NCoR–HDAC3 histone deacetylase complex[7,13]. The Polycomb system, which contributes to Xi silencing, is recruited by the RBP hnRNPK, which binds to the Xist B/C-repeat region[14,15]. Another RBP, the SPEN-related protein RBM15, was identified using both proteomic

and functional screening[8,10]. Follow-up studies have shown that RBM15 is an accessory subunit of the multiprotein complex that catalyzes RNA $N^6$-methyladenosine ($m^6A$)[16]. Other subunits of the $m^6A$ writer complex, particularly WTAP, were also identified in the Xist proteomic and functional screening experiments[8,10]. Notably, the recruitment of both RBM15 and WTAP to Xist depends on the Xist A-repeat region[8].

Building on initial observations implicating the $m^6A$ writer complex in XCI, it was reported that RBM15 directs $m^6A$ activity to two sites immediately downstream of the Xist A-repeat and E-repeat regions and perturbation of the complex, or of the protein YTHDC1, which binds to $m^6A$-modified RNA in the nucleus, strongly abrogates Xist-mediated gene silencing[16]. In contrast, other studies that analyzed Xist-mediated gene silencing reported minor or negligible effects on XCI following knockout of genes encoding subunits of the $m^6A$ writer complex[17] or following deletion of $m^6A$ sites in the vicinity of the Xist A-repeat[17,18]. Likely explanations for these discrepancies include the use of different cell models, different assays to assess Xist-mediated silencing and different strategies for perturbation of the $m^6A$ writer complex[19]. Of particular note is that $m^6A$ impacts the function of several thousand

[1]Developmental Epigenetics, Department of Biochemistry, University of Oxford, Oxford, UK. [2]Present address: Centre for Human Genetics, Nuffield Department of Medicine, University of Oxford, Oxford, UK. ✉e-mail: guifeng.wei@bioch.ox.ac.uk; neil.brockdorff@bioch.ox.ac.uk

mRNAs such that chronic or long-term gene knockout studies have the potential to lead to notable secondary or indirect effects.

In this study, we exploit an alternative approach, acute protein depletion with the dTAG degron system[20], to investigate the role of the m[6]A writer complex in Xist-mediated silencing. Our study demonstrates that the primary function of m[6]A on Xist RNA is to promote transcript turnover and that removal of METTL3 increases the rate of Xist-mediated silencing. Additionally, we find that the nuclear exosome targeting (NEXT) complex is essential for Xist degradation through a pathway that functions independently of the major nuclear m[6]A reader protein YTHDC1.

## Results

### Acute METTL3 depletion accelerates Xist-mediated gene silencing

In recent work, we analyzed the role of m[6]A in regulating nascent RNA processing, making use of the dTAG degron system[21] to rapidly deplete METTL3, the catalytic subunit of the m[6]A writer complex, in iXist-ChrX$_{Cast}$ (clone C7H8) mouse embryonic stem cells (mES cells)[17]. We used MeRIP-seq (m[6]A RNA immunoprecipitation and sequencing) to show that acute depletion of METTL3 for 2 h, followed by 24 h of Xist induction with continued METTL3 depletion, results in a rapid transcriptome-wide loss of m[6]A, including at major peaks in proximity to the A-repeat (exon 1) and E-repeat (exon 7) of Xist[21]. To further verify this finding, we performed an extended METTL3 depletion (24 h) before Xist induction (Extended Data Fig. 1a). Both conditions result in a near complete loss of the majority of m[6]A peaks across the transcriptome, including those on Xist (Extended Data Fig. 1b–e).

Here, we used the same degron strategy to investigate how acute loss of METTL3 and m[6]A affects Xist-mediated chromosome silencing. iXist-ChrX$_{Cast}$ mES cells have an interspecific *Mus castaneus* × 129 strain background with a stable XX karyotype and are engineered with a TetOn promoter for induction of Xist expression specifically from the *Mus castaneus* X chromosome[17]. Accordingly, we are able to isolate chromatin-associated RNA and subject it to sequencing (ChrRNA-seq) to accurately determine the relative expression level of active X (Xa) and Xi alleles for a large number of X-linked genes that have informative single-nucleotide polymorphisms (SNPs). In addition to two previously described lines with a METTL3 C-terminal FKBP12$^{F36V}$ tag[21], we derived two independent lines with METTL3 tagged with FKBP12$^{F36V}$ on the N terminus (Fig. 1a) in the iXist-ChrX$_{Cast}$ background. In all FKBP12$^{F36V}$-tagged cell lines METTL3 depletion occurred rapidly, within 2 h of adding the dTAG-13 reagent (Fig. 1b and Extended Data Fig. 2a), and moreover resulted in strongly reduced levels of METTL14, a subunit of the m[6]A complex that heterodimerizes with METTL3 (Fig. 1b and Extended Data Fig. 2b). We noted a reduction in levels of N-terminal FKBP12$^{F36V}$–METTL3, indicating that insertion of the tag affects translation or stability of METTL3 protein (Extended Data Fig. 2c).

We went on to quantify Xist-mediated silencing following METTL3 depletion as outlined in Fig. 1c. We analyzed silencing at an early time point, after 24 h Xist induction (with or without prior dTAG-13 treatment for 2 h), to minimize potential indirect effects of m[6]A depletion (Xist-mediated silencing in the iXist-ChrX$_{Cast}$ mES cells occurs progressively over a period of around 6 days[22]). The results are summarized in Fig. 1d. In the absence of dTAG-13 reagent, silencing levels in FKBP12$^{F36V}$-tagged lines were very similar to those seen in control (C7H8) cells, indicating that there are no effects attributable to addition of the degron at the C or N terminus of METTL3. Interestingly, addition of dTAG-13 resulted in a highly reproducible enhancement or acceleration of Xist-mediated silencing, evident in all four independently derived clones (Fig. 1d and Extended Data Fig. 1f). Of note, accelerated silencing is the converse of the silencing deficiency reported in prior work using constitutive knockout or knockdown of *METTL3* and other subunits of the m[6]A writer complex[16,17]. This difference likely reflects that acute METTL3 depletion is less influenced by indirect effects compared to chronic knockout experiments. Indeed, global gene expression differences in our study correlate better with m[6]A-modified mRNAs compared to published studies that used long-term knockout approaches[23] (Extended Data Fig. 2d). Principal component analysis (PCA) comparing silencing in untreated cells with acute depletion of METTL3 after 24 h of Xist induction (Fig. 1e), together with silencing analysis for previously defined gene categories[15,17,22] (Extended Data Fig. 2e–h), indicated that all X-linked genes are equally affected by accelerated silencing.

To determine whether the observed acceleration of Xist-mediated silencing is attributable to loss of METTL3 catalytic function, we performed complementation experiments using ectopic expression of *GFP–METTL3* transgenes. Transgene constructs were integrated under the control of the *Rosa26* constitutive promoter into the H5 clone with C-terminal degron-tagged METTL3 using CRISPR–Cas9-facilitated homologous recombination. In parallel, we established lines using a transgene encoding GFP–METTL3-D395A, a substitution that ablates METTL3 catalytic activity[24]. Constitutive expression of *GFP–METTL3* transgenes was maintained in both the presence and the absence of dTAG-13 (Extended Data Fig. 3a–c).

Both transgene encoded proteins reversed the observed reduction in METTL14 protein levels (Extended Data Fig. 3b,c versus Fig. 1b and Extended Data Fig. 2b), indicating the formation of stable GFP–METTL3/METTL14 heterodimers. Compared to wild-type (WT) GFP–METTL3, levels of GFP–METTL3-D395A were reduced (Extended Data Figs. 2c and 3b,c). This effect was not seen in the presence of dTAG-13 and is not linked to transcriptional levels (Extended Data Fig. 3b–d). A possible explanation is that METTL3-dependent m[6]A autoregulates the complementary DNA-derived ectopic *METTL3* transcript through RNA degradation. We went on to analyze Xist-mediated silencing in the transgene complementation lines. ChrRNA-seq analysis showed that ectopic expression of WT GFP–METTL3 fully complements the accelerated silencing phenotype observed following dTAG-13 treatment, whereas expression of GFP–METTL3-D395A has no effect (Fig. 1f and Extended Data Fig. 3e).

To confirm that accelerated Xi gene silencing is attributable to METTL3 catalytic activity, we made use of a recently developed

**Fig. 1 | Acute depletion of METTL3 results in accelerated XCI. a,** Schematic outline of N-terminal FKBP12$^{F36V}$ tagging of METTL3. **b,** Western blot showing the FKBP12$^{F36V}$–METTL3 fusion protein and METTL14 protein level for two independent clones (G4 and H1) after 2 or 24 h of dTAG-13 treatment. SETDB1 was used as a loading control. Short (middle) and long (bottom) exposure were also explored. **c,** Schematic outline detailing cell line specification and experimental design. **d,** Box plot showing the allelic ratio of X-linked genes (*n* = 396) from ChrRNA-seq analysis for each sample and condition indicated above and below, respectively. Two independent lines tagged with an N-terminal degron (G4 and H1 clone) and C-terminal FKBP12$^{F36V}$ (C3 and H5 clone) were included for this analysis, alongside untagged WT cells (C7H8 clone). The *y* axis denotes an allelic ratio ranging from 0 to 1. Two red dashed lines indicate the allelic ratio in ES cells (NoDox) and WT cells induced for 1 day with Dox.

*P* values were calculated using a two-sided paired *t*-test. **e,** PCA using allelic ratio of X-linked genes (*n* = 374) for samples in **d**, along with time-course WT (C7H8) samples. **f,** Box plot depicting the allelic ratio of X-linked genes (*n* = 410) from ChrRNA-seq analysis for the complementation assay where either GFP–METTL3 (WT_P2B3 clone) or GFP–METTL3-D395A (D395A_1F clone) was expressed from the *Rosa26* locus in a C-terminal METTL3 dTAG degron cell line (H5). Both WT_P2B3 and D395A_1F clones retain both X$_{cast}$ and X$_{129}$. The red dashed line indicates the allelic ratio at 0.5. Samples and conditions are indicated above and below, respectively. *P* values were calculated using a two-sided paired *t*-test. Two biological replicates were averaged. In box plots (**d,f**), center lines indicate the median, box limits indicate the first and third quartiles and whiskers indicate 1.5× the interquartile range (IQR).

pharmacological METTL3 inhibitor, STM2457 (ref. 25). STM2457 treatment of parental C7H8 XX mES cells resulted in changed levels of YTHDC1 and WTAP and altered ratios of alternative splicing (Extended Data Fig. 4a,b), as occurs following acute depletion of METTL3 (ref. 21). We then analyzed Xi gene silencing following treatment of cells with either DMSO or STM2457 for 6 h, followed by 24 h of Xist induction under continued treatment (Extended Data Fig. 4c). ChrRNA-seq analysis revealed that low-dose STM2457 treatment results in a modest increase in Xist RNA levels and accelerated XCI dynamics, with higher doses eliciting a clearly enhanced effect (Extended Data Fig. 4d,e).

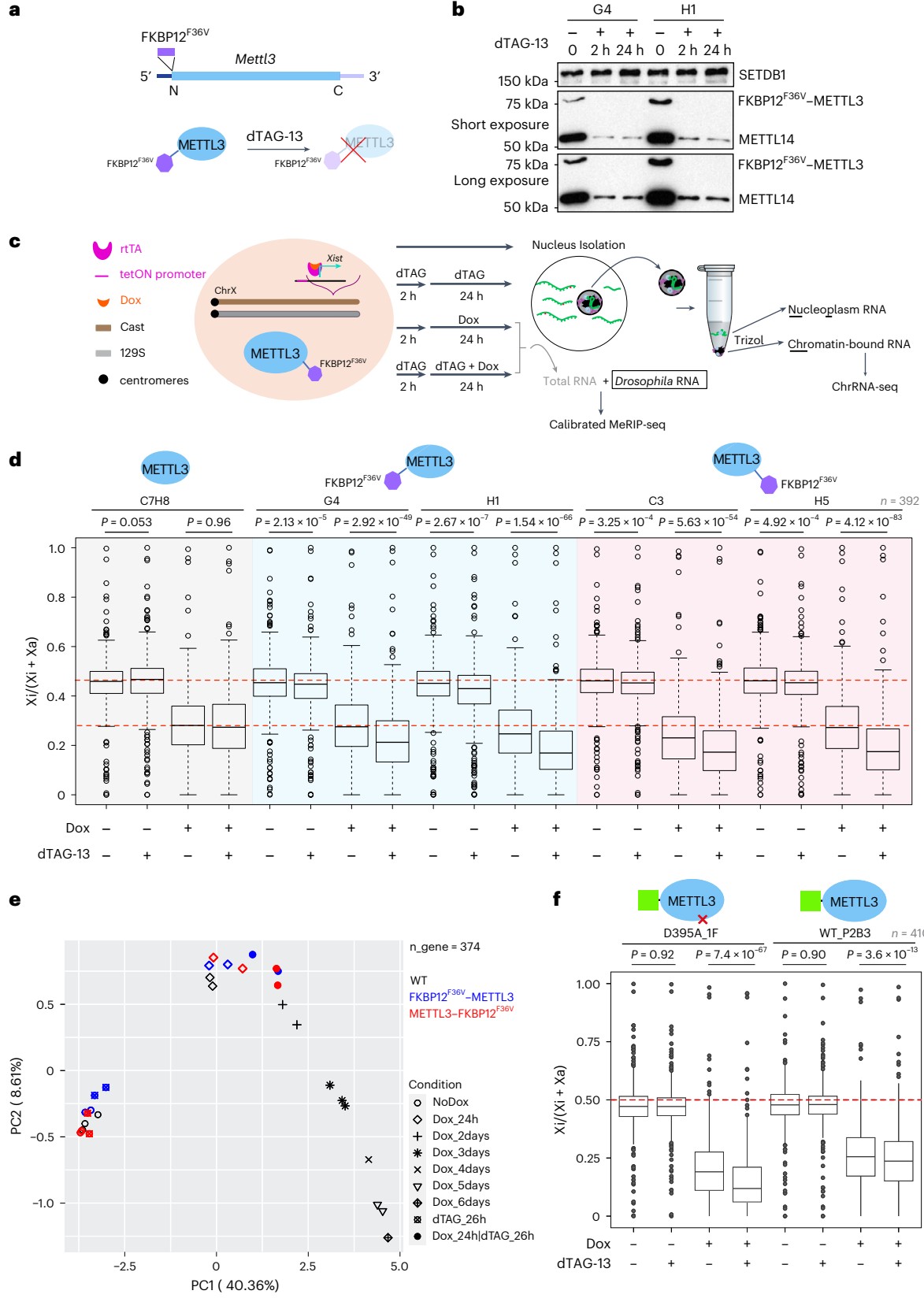

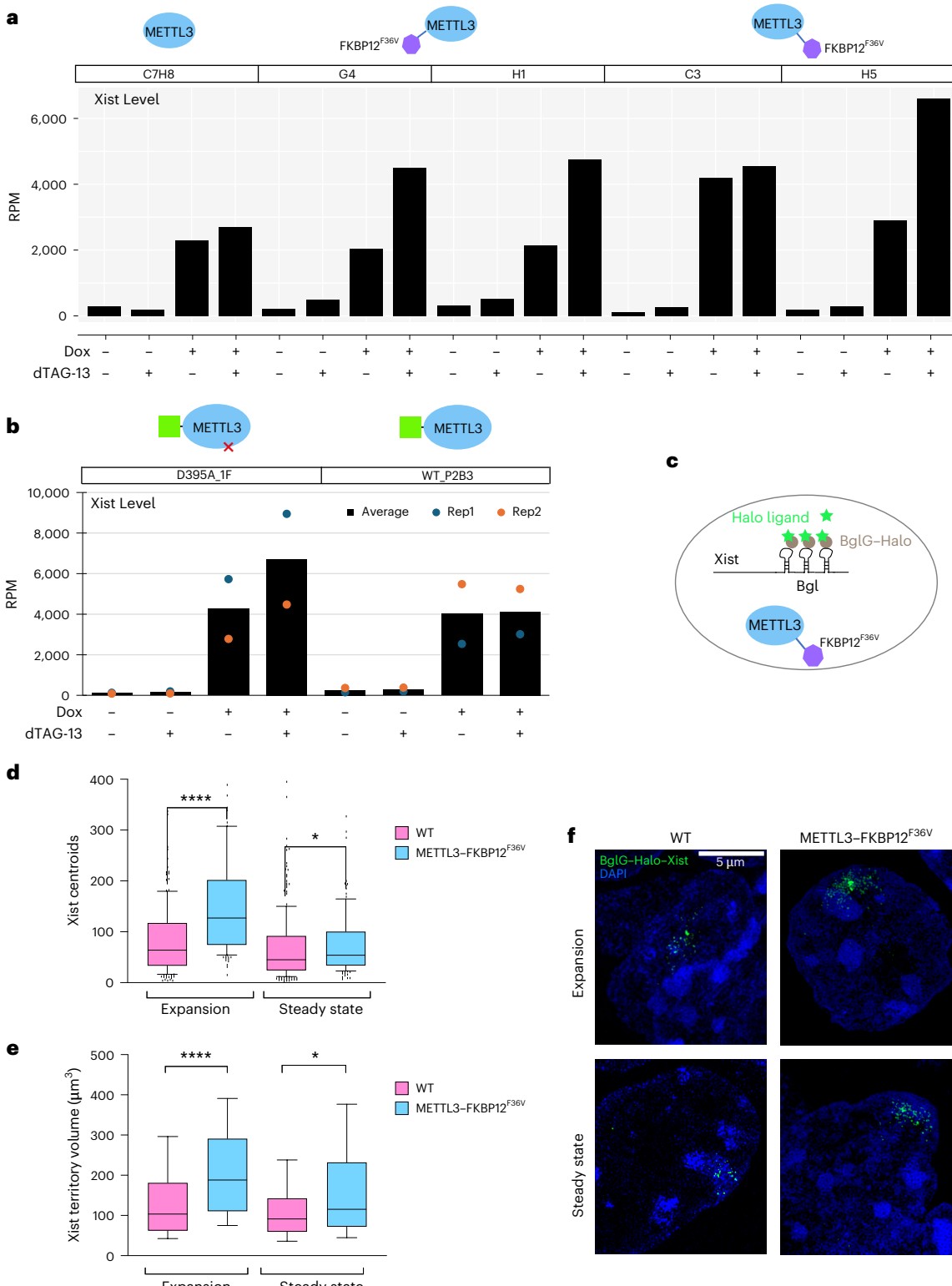

**Fig. 2 | Xist RNA levels are elevated following acute depletion of METTL3. a**, Bar plot showing the expression level (RPM; reads per million mapped reads) of Xist from ChrRNA-seq analysis for each sample and condition described in Fig. 1d. Each bar represents one or two biological replicates. **b**, Expression level of Xist, as in **a**, from ChrRNA-seq analysis for samples and conditions described in Fig. 1f. Two biological replicates were averaged. **c**, Schematic of cell line with METTL3 dTAG degron and Xist exon 7 18× Bgl stem-loop bound by BglG–Halo fusion protein for fluorescent imaging of Xist RNA. **d,e**, Box plots showing the number of Xist molecules (**d**) and Xist cloud volume (**e**) analyzed by RNA-SPLIT for both

WT (pink; data from a previous study[26]) and METTL3 dTAG-depleted cells (blue) at both the expansion phase (1.5 h of Xist induction) and steady-state phase (24 h of Xist induction), with $n > 123$ cells. Significance was determined using a two-tailed Mann–Whitney test (*$P = 0.0135$ in **d** and *$P = 0.0194$ in **e**; ****$P < 0.0001$). Center lines indicate the median, box limits indicate the first and third quartiles and whiskers indicate 1.5× the IQR. **f**, Representative 3D-SIM images (maximum projection) of Xist RNA (HaloTag, green) in WT (data from a previous study[26]) and METTL3–FKBP12[F36V] cells at both expansion and steady-state phases. DNA is counterstained with DAPI (blue).

Collectively, these results confirm the role of METTL3 in regulating the rate of Xist-mediated silencing and further support that METTL3 function in this scenario is through m6A catalysis.

## Accelerated XCI is linked to increased levels and stability of Xist RNA

We noted a close correlation between accelerated Xist-mediated silencing and levels of Xist RNA, as determined from ChrRNA-seq datasets. Specifically, increased Xist RNA levels up to approximately twofold were apparent following acute METTL3 depletion in the independent N-terminally and C-terminally tagged cell lines (Fig. 2a and Extended Data Fig. 1f). Additionally, complementation with WT GFP–METTL3 but not GFP–METTL3-D395A restored Xist RNA levels to those seen in untreated cells (Fig. 2b and Extended Data Fig. 3f).

To further investigate the effect of acute METTL3 depletion on Xist RNA, we used super-resolution three-dimensional structured illumination microscopy (3D-SIM) imaging to assay features of Xist RNA domains at the single-cell level. For these experiments, we engineered the METTL3 C-terminal degron into previously described iXist-ChrX$_{129}$ XX mES cells in which the TetOn-inducible Xist allele has a BglI stem-loop array integrated into Xist exon 7, allowing detection of Xist RNA molecules through binding of a BglG–HaloTag fusion protein labeled with fluorescent Halo dyes[26] (Fig. 2c and Extended Data Fig. 2a). In prior work using this system, we reported that Xist RNA accumulates to maximal levels of around 50–100 molecules per cell over a period of 1.5–5 h, referred to as the expansion phase. A time point of 24 h was previously selected where Xist RNA levels have attained a steady state. Using these parameters, we observed an increase in the number of Xist molecules following acute METTL3 depletion (Fig. 2d,f), in agreement with the ChrRNA-seq data. In addition, the volume encompassing Xist centroids (overall Xist domain size) was also significantly increased at 1.5 h (expansion phase) and 24 h (steady-state phase) of Xist induction (Fig. 2e,f). These observations confirm that acute depletion of METTL3 and m6A leads to increased Xist RNA levels and an enlarged Xist domains corresponding to the Xi territory.

We also investigated whether other well-characterized nuclear long noncoding RNAs (lncRNAs) that are m6A modified are similarly affected by acute METTL3 depletion. Accordingly we examined levels of Neat1, Malat1 and Kcnq1ot1 RNAs, all of which are expressed in mES cells and have high levels of METTL3-dependent m6A (Extended Data Fig. 5a–c). Levels of Neat1 and Malat1 were unaffected by acute METTL3 depletion; however, similarly to Xist RNA, Kcnq1ot1 levels increased approximately 1.5–2-fold (Extended Data Fig. 5d–f). The effect on Kcnq1ot1 levels was dependent on METTL3 catalytic activity (Extended Data Fig. 5f, right).

Increased abundance of Xist RNA following acute depletion of METTL3 could result from changes in the rate of Xist RNA transcription and/or RNA turnover. To investigate these possibilities, we applied RNA-SPLIT (sequential pulse localization imaging over time) coupled to super-resolution 3D-SIM microscopy[26] to differentially label successive waves of Xist transcripts (presynthesized and newly synthesized) before fixation. The labeling regimen for determining turnover rates is shown in Fig. 3a. Experiments were performed at both the expansion phase and the steady-state phase using a 20-min interval. A 2-h dTAG-13 treatment was performed before Xist induction to minimize secondary or indirect effects. Example images in Fig. 3b are from the expansion phase. As shown previously, turnover of Xist RNA occurs within 140 min at the expansion phase and 220 min at the steady-state phase[26] (Fig. 3b,c). In marked contrast, following acute METTL3 depletion, there was little Xist RNA turnover detectable across the entire time course of the experiment (220 min) during both the expansion and the steady-state phases (Fig. 3b,c and Supplementary Fig. 1). Reduced turnover of Xist transcripts was also demonstrated using an orthogonal approach, SLAM-seq[27], based on transient 4sU labeling of newly synthesized RNA (Extended Data Fig. 6a–c). Allele-specific analysis using these

sequencing data confirmed accelerated silencing and increased Xist RNA levels following acute METTL3 depletion (Extended Data Fig. 7).

We further applied RNA-SPLIT to measure Xist transcription rates in the presence and absence of m6A, achieved by quantifying foci for newly synthesized Xist RNA over time during expansion phase. As shown in Fig. 3d,e, there is a significantly reduced transcription rate in the acute METTL3 depletion condition compared to WT cells. This finding is consistent with prior work demonstrating a feedback mechanism that links Xist transcription and turnover[26]. Accordingly, we conclude that loss of m6A results in accelerated X-chromosome silencing because of overaccumulation of Xist transcripts.

## Xist RNA turnover is mediated by the ZCCHC8–NEXT complex independently of YTHDC1

The cellular functions of m6A are mediated by reader proteins that can bridge to downstream pathways. YTHDC1 is the best-characterized protein that directly recognizes m6A in the nucleus[28]. Interestingly, YTHDC1 coimmunoprecipitation experiments revealed an association with ZCCHC8, a core subunit of the NEXT complex that targets nonpolyadenylated transcripts in the nucleus for degradation[29]. Both YTHDC1 and ZCCHC8 have a role in degrading nonpolyadenylated chromatin-associated regulatory RNAs (carRNAs), for example, PROMPTs and eRNAs[30,31]. Consistent with this finding, ZCCHC8 has been reported to interact with YTHDC1 in experiments using stable isotope labeling in cell culture and mass spectrometry[29]. The YTHDC1–RNA exosome axis has also been implicated in the degradation of other nuclear RNAs, for example, SμGLT lncRNA[32] and *C9ORF72* repeat RNA[33].

To investigate whether YTHDC1 is important for regulating Xist RNA turnover, we used CRISPR–Cas9 facilitated genome editing to establish XX mES cell-derived lines with the FKBP12$^{F36V}$ degron tag inserted into the gene encoding YTHDC1 (Fig. 4a). YTHDC1 depletion on addition of dTAG-13 reagent was validated by western blot analysis (Fig. 4b,c) and examination of the effects on *Tor1aip2* alternative last exon splicing, which is significantly affected by METTL3 and m6A and conditional *YTHDC1* knockout[21] (Extended Data Fig. 8a,b). We went on to assay Xist-mediated silencing and Xist RNA levels following YTHDC1 depletion as described above. We observed no increase in the silencing rate or in the levels of Xist RNA in two independent degron-tagged cell lines (Fig. 4d,e and Extended Data Fig. 8c,d). If anything, both Xist RNA levels and the silencing efficiency of X-linked genes appeared modestly reduced compared to controls. However, this reduction was notably less pronounced than that observed with acute depletion of known XCI regulators, for example PCGF3/5 (ref. 22) (Fig. 4d). Additionally, there was little or no effect on Xist RNA stability by acute depletion of YTHDC1, as determined by SLAM-seq (Extended Data Fig. 6c). Similarly, Kcnq1ot1 RNA, levels of which increase following METTL3 depletion, were unaffected by YTHDC1 depletion (Extended Data Fig. 5g–i, left).

We went on to investigate the role of the NEXT complex in Xist RNA turnover by inserting the FKBP12$^{F36V}$ degron tag into the gene encoding the core subunit ZCCHC8 in XX mES cells (Fig. 5a and Extended Data Fig. 9a,b). As an additional control, we established XX mES cell lines with the FKBP12$^{F36V}$ degron tag inserted into the gene encoding ZFC3H1, an essential subunit of the poly(A) tail exosome targeting (PAXT) complex[34] (Fig. 5b and Extended Data Fig. 9a,c). PAXT mediates an alternate pathway for degradation of polyadenylated RNA in the nucleus, potentially functioning as a timer to remove aberrant RNAs that are not efficiently exported[34]. dTAG-13 treatment led to rapid and complete depletion of the FKBP12$^{F36V}$-tagged proteins within 2 h (Fig. 5a,b). We noted that m6A-dependent alternative last exon splicing of the *Tor1aip2* gene was not affected by acute depletion of ZCCHC8 or ZFC3H1, indicating that neither NEXT nor PAXT is required for m6A deposition on target mRNAs (Extended Data Fig. 9d,e). Acute ZCCHC8 depletion was further validated by monitoring upregulation of PROMPTs (Extended Data Fig. 10a,b).

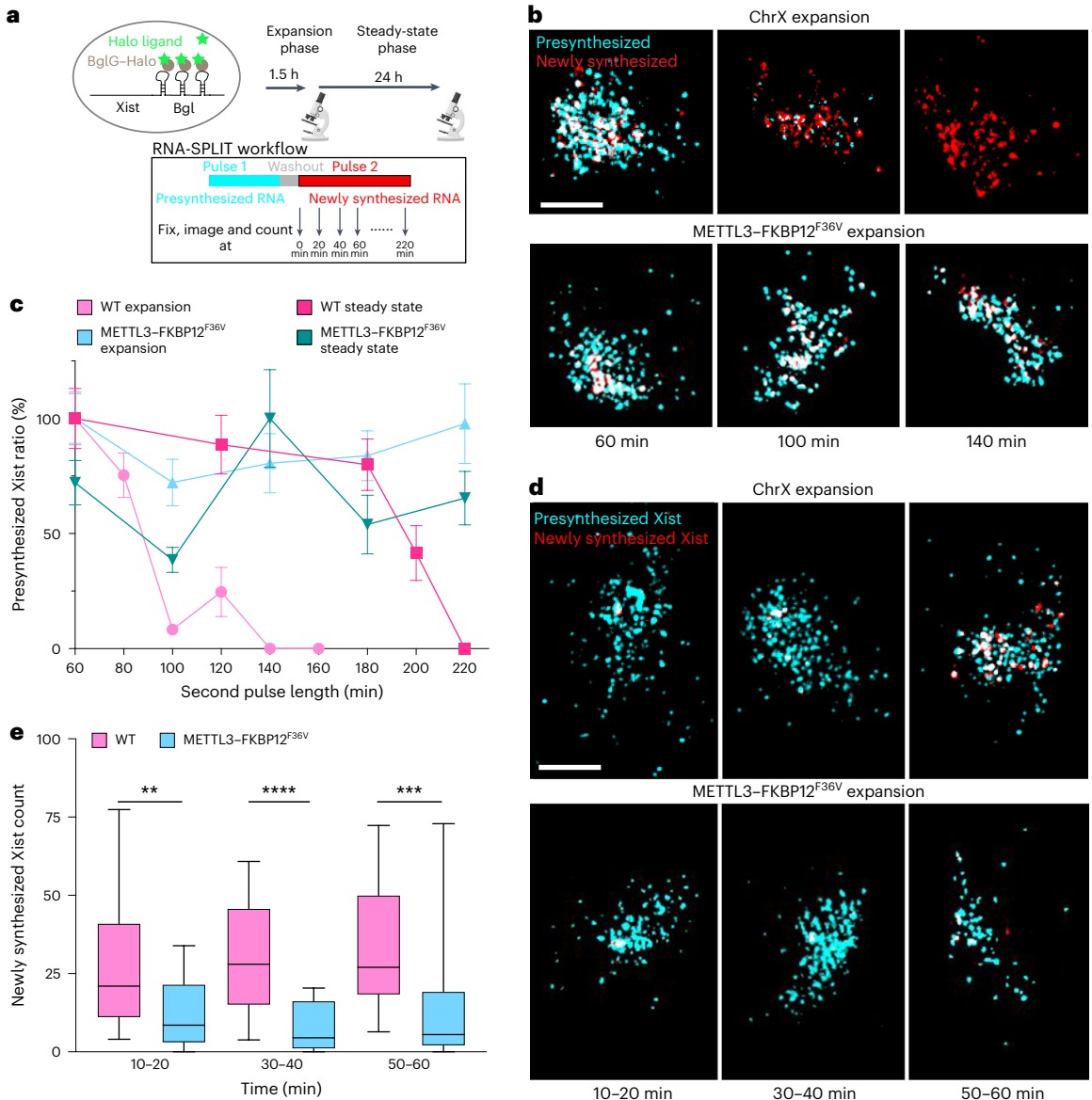

**Fig. 3 | Acute depletion of METTL3 stabilizes Xist RNA. a**, Schematic detailing RNA-SPLIT to assess the rate of Xist RNA turnover. **b**, Representative 3D-SIM images illustrating *z*-project of presynthesized (cyan) and newly synthesized (red) Xist molecules during expansion phase for WT (top; data from a previous study[26]) and METTL3 dTAG cells (bottom). Images are representative of 60, 100 and 140 min of pulse 2 labeling. Scale bar, 2 μm. **c**, Plot showing quantification of Xist RNA turnover during expansion (pink and light blue) and steady state (magenta and turquoise) for WT (pink and magenta; data from a previous study[26]), METTL3 dTAG (light blue and turquoise), with *n* > 20 cells per time

point. Error bars represent the s.e.m. and the centers indicate the mean. **d**, As in **b** but images are representative of pulse 2 labeling at indicated times. Scale bar, 2 μm. **e**, Box plot showing Xist RNA transcription over time during expansion phase for WT (pink; data from a previous study[26]) and dTAG-13-treated METTL3 degron cells (blue), with *n* > 22 cells per time point. Significance was determined using a two-tailed Mann–Whitney test (**P* = 0.0091, ****P* = 0.008 and *****P* < 0.0001). Center lines indicate the median, box limits indicate the first and third quartiles and whiskers indicate 1.5× the IQR.

We went on to perform ChrRNA-seq to assay Xist-mediated silencing after 24 h of Xist induction, in either the presence or the absence of NEXT or PAXT complexes using the approach described above for analysis of METTL3 and YTHDC1. As shown in Fig. 5c, acute depletion of ZCCHC8 resulted in strongly accelerated silencing, whereas depletion of ZFC3H1 resulted in a marginal reduction in gene silencing (Fig. 5d). Consistent with these observations, Xist RNA levels were elevated following depletion of ZCCHC8 but not of ZFC3H1, where Xist levels were slightly lower if anything (consistent with marginally reduced gene silencing) (Fig. 5e,f). The higher Xist RNA levels and stability following acute ZCCHC8 depletion are correlated with an even more marked acceleration of silencing than that was seen with acute METTL3 depletion (Fig. 1d versus Fig. 5c).

Upregulation of Xist RNA upon acute depletion of ZCCHC8 and NEXT is evident across the entire transcript (Extended Data Fig. 10c). PCA indicates that accelerated silencing affects X-linked genes equivalently across the X chromosome, which contrasts with the silencing deficiency observed upon knockout of the key silencing factor SPEN or deletion of B/C-repeats in Xist (Extended Data Fig. 10d). Levels of Kcnq1ot1 RNA were similarly elevated following acute depletion of ZCCHC8 but not ZFC3H1 (Extended Data Fig. 5g–i, right). SLAM-seq analysis indicates that elevated Xist RNA levels following acute ZCCHC8 depletion are attributable to increased transcript stability (Extended Data Fig. 6c). Taken together these results suggest that m⁶A promotes Xist RNA turnover by the NEXT complex independently of YTHDC1.

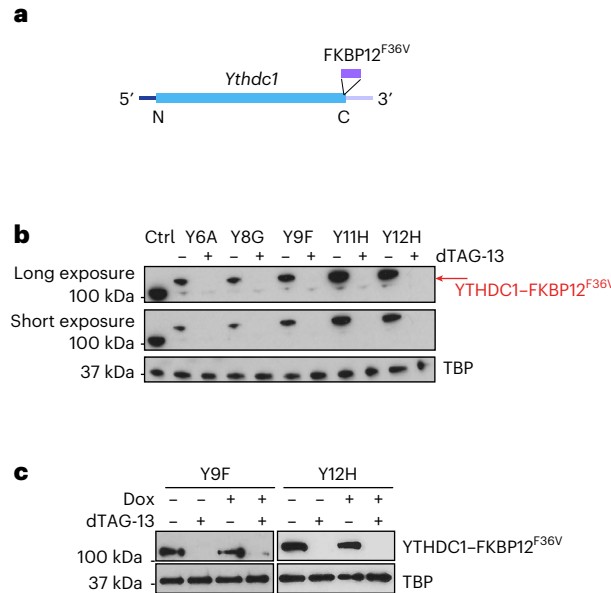

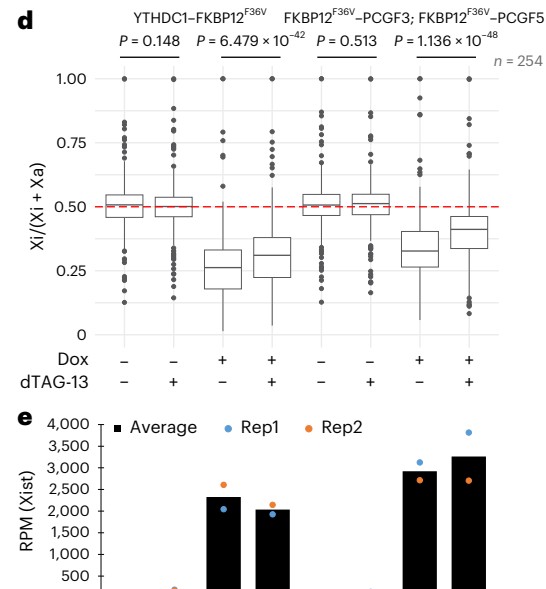

**Fig. 4 | Xist RNA turnover is independent of the m⁶A nuclear reader protein YTHDC1. a**, Strategy for in-frame insertion of FKBP12^F36V into YTHDC1. **b**, Western blot showing the protein size and level of YTHDC1 upon 0 or 2 h of dTAG-13 treatment. The tagged protein is indicated by the red arrow. TBP was used as a loading control. The blot is representative of three biologically independent experiments. **c**, Western blot showing the protein level of YTHDC1–FKBP12^F36V for ChrRNA-seq samples shown in **d**. TBP was used as a loading control. **d**, Box plot showing the allelic ratio of X-linked genes (*n* = 263) from ChrRNA-seq analysis for YTHDC1 dTAG degron (left 4 samples) and published data on PCGF3/5 degron

(right 4 samples) mES cells. The experimental design is as described in Fig. 1c. The red dashed line indicates allelic ratio at 0.5. Samples and conditions are indicated above and below, respectively. *P* values, as indicated, were calculated using a two-sided paired *t*-test. Two replicates were averaged. Center lines indicate the median, box limits indicate the first and third quartiles and whiskers indicate 1.5× the IQR. **e**, Bar plot showing the expression level of Xist from ChrRNA-seq analysis for samples and conditions described in **d**, with individual replicates represented as dots.

## Discussion

The acceleration of Xist-mediated silencing that we observe following acute METTL3 depletion conflicts with prior studies that used long-term knockout or knockdown strategies and reported abrogation of silencing to a greater or lesser degree[16,17]. Differences observed in our experiments are likely attributable to reduced indirect effects from acute as opposed to chronic depletion of METTL3. This conclusion is supported by the observation that global mRNA expression changes show an improved correlation with m⁶A target mRNAs with acute METTL3 depletion compared to chronic knockout or knockdown. Although we cannot pinpoint with certainty why perturbation of the m⁶A pathway in prior studies resulted in abrogated Xist-mediated silencing, a possible explanation could lie in changed levels of mRNAs encoding proteins that regulate expression of key silencing factors. Indeed, mRNAs encoding the silencing factors SPEN, HDAC3 and HNRNPU are all m⁶A modified. Regardless, our findings underscore that caution needs to be exercised in interpreting experiments involving chronic perturbation of the m⁶A pathway because of its roles in regulating transcription and RNA metabolism at a global level[35,36]. Intriguingly, the Xist A-repeat region is bound by RBPs such as SPEN, which promotes Xist RNA stability, and RBM15 and WTAP, which contribute to its destabilization through m⁶A modification. The opposing effects of these pathways on Xist RNA stability highlight a regulatory balance that warrants further investigation.

Our previous work reported that deletion of the m⁶A region in Xist exon 1 has minimal effects on Xist RNA levels and XCI dynamics[17,18]. In this study, we generated a deletion of the m⁶A region within Xist exon 7, either alone or in combination with deletion of the exon 1 m⁶A region, to further investigate the functional contribution of distinct m⁶A regions to Xist RNA regulation. Unexpectedly, deletion of the exon 7 m⁶A region resulted in impaired gene silencing and a marked reduction in Xist RNA levels (Supplementary Fig. 2). Given that the deletion is located far from the Xist promoter, we speculate that the observed reduction in

RNA levels is likely because of decreased transcript stability. Notably, the double-deletion line displayed a similar phenotype to the exon 7 m⁶A region single deletion, with no evidence of additive effects (Supplementary Fig. 2). These findings can be interpreted in several ways. The exon 7 m⁶A region may overlap with an important RNA stability element, the activity of which is inhibited by m⁶A deposition. Alternatively, the deletion may impair the activity of a neighboring functional element, such as the Xist E-repeat, which is known to influence Xist RNA localization, or may alter the splicing pattern of the transcript. Lastly, the deletion could lead to structural rearrangements in the RNA by juxtaposing sequences that are normally spatially separated, thereby disrupting higher-order folding of Xist RNA and compromising its stability or function. Disentangling the precise roles of individual m⁶A sites within Xist remains a notable challenge and will require further targeted investigation.

Our results indicate that accelerated silencing following acute METTL3 depletion is linked to increased levels of Xist RNA, which in turn result from decreased Xist RNA turnover. This interpretation is supported by the similar phenotype seen following acute depletion of the NEXT subunit ZCCHC8. We envisage that higher Xist RNA levels result in an increased number of molecules accumulating on the X chromosome, thereby amplifying the recruitment of silencing factors that mediate gene silencing in X inactivation. Of note, the increased Xist RNA levels linked to reduced turnover are tempered by a reduction of Xist transcription rates, consistent with our prior analyses indicating feedback control between Xist transcription and stability[26]. The basis for feedback control between Xist transcription and turnover remains unknown and is an important topic for future studies.

The finding that NEXT but not PAXT regulates Xist RNA turnover is perhaps unexpected given that Xist is polyadenylated and NEXT has been linked to regulating nonpolyadenylated nuclear RNAs such as PROMPTs and eRNAs[37]. One possible explanation is that the poly(A) tail of Xist molecules localized along the Xi chromosome is protected

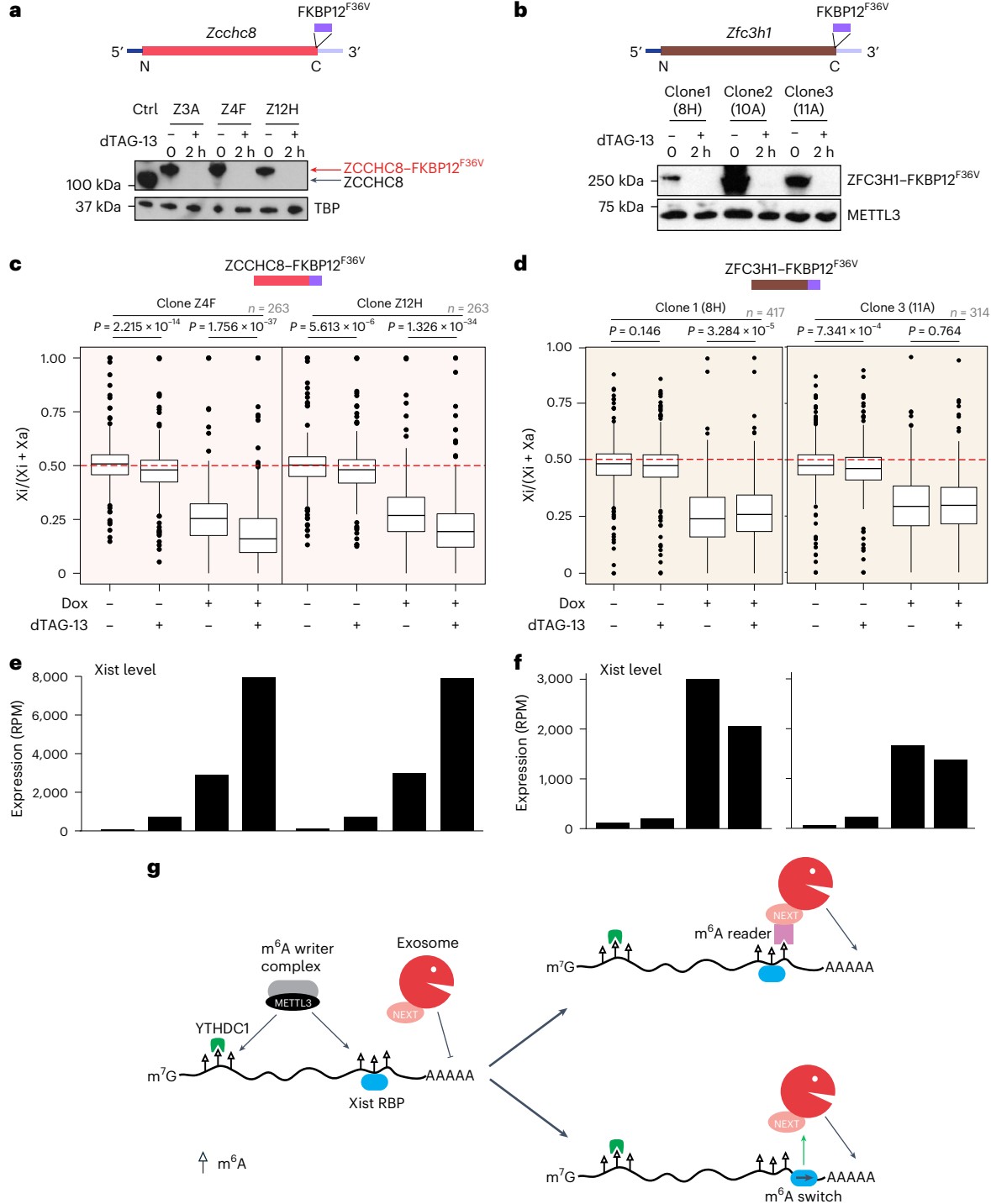

**Fig. 5 | The NEXT complex mediates Xist RNA turnover in the cell nucleus.**
**a**, Top, strategy showing in-frame insertion of FKBP12^F36V into ZCCHC8. Bottom, western blot showing the protein level of ZCCHC8–FKBP12^F36V after 0 or 2 h of dTAG-13 treatment. TBP was used as a loading control. The red and black arrows indicate the protein size for ZCCHC8–FKBP12^F36V and untagged ZCCHC8. **b**, As in **a** but for ZFC3H1. METTL3 was used as a loading control. **c**, Box plot showing the allelic ratio of X-linked genes from ChrRNA-seq analysis for ZCCHC8 dTAG degron samples. The experimental design is as described in Fig. 1c. The red dashed line indicates the allelic ratio at 0.5. Samples (two independent clones, Z4F and Z12H) and conditions are indicated above and below, respectively.

$P$ values, as indicated, were calculated using a two-sided paired $t$-test. **d**, As in **c** but for ZFC3H1. Note that only the proximal 138 MB of X chromosome in the 11A clone is informative for allelic-speific analysis. In box plots (**c**,**d**), center lines indicate the median, box limits indicate the first and third quartiles and whiskers indicate 1.5× the IQR. **e**, Bar plot showing the expression level of Xist from ChrRNA-seq analysis for samples and conditions described in **c**. **f**, As in **e** but for ZFC3H1 clones described in **d**. Each bar in **e**,**f** represents one biological replicate. **g**, Schematic depicting alternative models for m6A-mediated regulation of Xist RNA turnover as discussed in the main text. m6A sites are indicated as lollipops with open triangle.

from NEXT activity, for example, by poly(A)-binding proteins such as PABPN1 or by a folded back structure similar to the lncRNA MALAT1 (ref. 38). Progressive erosion of this protection over time would, thus,

initiate NEXT–exosome complex engagement to trigger degradation from the 3′ end. Consistent with this idea, our previous RNA-SPLIT analysis demonstrates that Xist molecules in WT cells remain relatively

stable for an extended period of around 90 min before showing rapid exponential degradation kinetics[26].

The marked increase in Xist RNA stability, Xist RNA levels and Xi silencing rate observed with both METTL3 and ZCCHC8 depletion, suggests that m6A and the NEXT complex function together to regulate Xist RNA turnover. On the basis of prior studies identifying the YTHDC1–NEXT axis and its role in regulating carRNAs, we anticipated that YTHDC1 could bridge Xist RNA with NEXT for m6A-mediated control of turnover. However, given that acute YTHDC1 depletion has little to no notable effect on Xist RNA turnover, we speculate that there is an alternative pathway that links m6A to the NEXT complex. In support of this idea, we observe no effect of YTHDC1 depletion on levels of the lncRNA Kcnq1ot1 (Extended Data Fig. 5i), which, like Xist, is chromatin associated and functions to silence genes in cis[39]. We envisage two possible models, a direct pathway whereby an unidentified reader protein interacts with both m6A and NEXT or an indirect pathway where m6A enhances the association of Xist RBPs that consequently promote NEXT–exosome complex access and activity on Xist (Fig. 5g). In relation to the latter possibility, hnRNPC and hnRNPA2B1 proteins that have been previously reported to function indirectly through an 'm6A switch' mechanism[40,41] were identified in mass spectrometry experiments as Xist RNA interactors[8]. Together with previous reports showing that m6A can promote nuclear RNA stability through YTHDC1 (refs. 42,43), our findings underscore the complexity and context-dependent nature of m6A function in regulating nuclear RNA stability. Additional studies are required to fully elucidate the factors and pathways governing m6A-mediated nuclear RNA stability.

In conclusion, we find that the m6A pathway functions in conjunction with the NEXT complex to modulate Xist RNA turnover and thereby contributes to the feedback control mechanisms that determine Xist RNA levels during normal development.

## Online content

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

## Methods

### Tissue culture

All mES cells were grown in feeder-dependent conditions on gelatinized plates at 37 °C in a 5% $CO_2$ incubator. Mitomycin C-inactivated mouse fibroblasts were used as feeders. mES cell medium consisted of DMEM (Thermo Fisher) supplemented with 10% fetal calf serum (Merck), 2 mM L-glutamine (Thermo Fisher), 1× nonessential amino acids (Thermo Fisher), 50 µM β-mercaptoethanol (Thermo Fisher), 50 g ml$^{-1}$ penicillin–streptomycin (Thermo Fisher) and 1 ml of medium conditioned with leukemia-inhibitory factor made in house. Xist expression was induced by the addition of 1 µg ml$^{-1}$ doxycycline (Dox) (Sigma-Aldrich, D9891) for 24 h. Protein of interest with *FKBP12*[F36V] knock-in was trigged for degradation by the addition of dTAG-13 (100 nM) (gift from J. Bradner lab).

### Molecular cloning and CRISPR–Cas9-mediated knock-in

We followed an established strategy for *FKBP12*[F36V] knock-in[21]. Briefly, single guide RNA (sgRNA) targeting near the *YTHDC1* or *ZCCHC8* stop codon was designed by online tool CRISPOR[44] (http://crispor.tefor.net/); oligos were synthesized by Invitrogen and then cloned into pSpCas9(BB)-2A-Puro (PX459) V2.0 (Addgene, 62988) backbone following the instructions. A donor vector was built by Gibson assembly (New England Biolabs) of homology arms (~400 bp) PCR-amplified from genomic DNA and the *FKBP12*[F36V] sequence amplified from plasmid pLEX_305-N-dTAG (Addgene, 91797)[20] or other coding sequences (for example, GFP–METTL3) described below. To achieve site-specific mutagenesis from the plasmid, we followed the protocol from the QuikChange Lightning site-directed mutagenesis kit (Agilent) or Gibson assembly. This strategy was used for generating the mES cell line with GFP–METTL3 or GFP–METTL3-D395A at the *Rosa26* locus, as described below. The Cas9 sgRNA-containing plasmid and donor vector were cotransfected at a molar ratio of 1:6 into XX mES cells on a six-well plate using Lipofectamine 2000 according to the manufacturer's protocol (Thermo Fisher). Transfected cells were passaged at different densities into three Petri dishes with feeders, 24 h after transfection. The next day, cells were subjected to puromycin selection (4.5 µg ml$^{-1}$) for 48 h and then grown in regular mES cell medium until mES cell colonies were ready to be picked and expanded. Western blot analysis was used to screen colonies for *FKBP12*[F36V] knock-in as this results in slower mobility of the FKBP12[F36V]-fused protein compared to the WT protein in an SDS–PAGE gel and disappearance of the WT band on western blot. Selected clones were further characterized by PCR of genomic DNA and Sanger sequencing to confirm correct knock-in and homozygosity. The karyotype status of the X chromosome was checked by PCR. The sensitivity of selected clones to dTAG-13 treatment was validated by western blot. A summary of clone names can be found in Supplementary Table 1. Primers used in this study are listed in Supplementary Table 2.

### Complementation using *METTL3* or catalytic mutant *METTL3* transgenes in the *Rosa26* locus

CRISPR–Cas9-mediated knock-in was used to integrate *METTL3* or catalytic mutant *METTL3*[D395A] (ref. 24) transgenes into the *Rosa26* locus. To distinguish transgene from endogenous METTL3 and FKBP12[F36V]-tagged METTL3, we added a sequence encoding an in-frame N-terminal GFP tag. sgRNA (CGCCCATCTTCTAGAAAGAC) was used for targeted knock-in. METTL3-D395A was constructed by Gibson assembly. PCR and western blot were used for screening colonies. The expression and nuclear localization of the GFP–METTL3 fusion proteins were confirmed by fluorescence microscopy. Elimination of the $X_{129}$ chromosome occurred in several clones, as indicated in the figure legends. This did not preclude assaying relative silencing efficiency and Xist RNA levels in the presence and absence of Dox and dTAG-13.

### Western blot

Total cell lysates or quantified cell fractionations were resolved on a polyacrylamide gel and transferred onto PVDF or nitrocellulose membrane by quick transfer. Membranes were blocked by incubating them for 1 h at room temperature in 10 ml of 5% w/v Marvell milk powder. Blots were incubated overnight at 4 °C with the primary antibody, washed four times for 10 min with PBST and incubated for 1 h with secondary antibody conjugated to horseradish peroxidase (HRP). After washing five times for 5 min with PBST, bands were visualized using enhanced chemiluminescence (GE Healthcare). TBP, SETDB1 and KAP1 served as loading controls. Antibodies used in this study were anti-METTL3 (Abcam, ab195352), anti-METTL14 (Sigma-Aldrich, HPA038002), anti-RBM15 (Proteintech, 10587-1-AP), anti-YTHDC1 (Sigma-Aldrich, HPA036462), anti-TBP (Abcam, ab51841), anti-WTAP (Proteintech, 10200-1-AP), anti-SETDB1 (Proteintech, 11231-1-AP), anti-ZCCHC8 (Proteintech, 23374-1-AP), anti-ZFC3H1 (Sigma-Aldrich, HPA007151) and anti-KAP1 (Abcam, ab10484), together with anti-rabbit IgG HRP donkey (Amersham, NA934V) and anti-mouse IgG HRP sheep (Amersham, NXA931V). Primary antibodies used for western blot were diluted at 1:1,000 whilst secondary antibodies were diluted at 1:2,000.

### RNA-SPLIT

RNA-SPLIT was carried out as detailed previously[26]. Briefly, mES cells were grown on gelatin-coated 18 × 18-mm no. 1.5H precision coverslips (±5 µm tol.; Marienfeld Superior) in a six-well plate on a layer of feeder cells. When the mES cells reached 60–70% confluency, Xist expression was induced for 1.5 or 24 h using 1 µg ml$^{-1}$ Dox. Cells were then incubated with 50 nM diAcFAM HaloTag ligand (488 nm, Promega) and 1 µg ml$^{-1}$ Dox for 45 min, before being washed with ES cell medium containing 1 µg ml$^{-1}$ Dox for 15 min. Next, different coverslips were incubated with 50 nM JF-585 HaloTag ligand (kindly provided by L. Lavis, Howard Hughes Medical Institute Janelia) and 1 µg ml$^{-1}$ Dox for different times to label newly synthesized Xist RNA molecules, (0, 60, 80, 100, 120, 140, 160, 180, 200 and 220 min to assess Xist turnover or 10, 20, 30, 40, 50 and 60 min to assess Xist RNA transcription dynamics) before being washed with PBS. Cells were fixed for 10 min at room temperature with 2% formaldehyde prepared fresh in PBS (pH 7) before a stepwise exchange to PBST (0.05% Tween-20) and permeabilization with 0.2% Triton X-100 for 10 min, followed by two washes with PBST. Subsequently, cells were incubated with 2 µg ml$^{-1}$ DAPI in PBST for 10 min, before being washed briefly with PBS followed by double-distilled $H_2O$. Cells were then mounted centrally on the unfrosted side of Superfrost Plus microscopy slides (VWR) using Vectashield soft mount medium, sealed with clear nail polish and imaged using the DeltaVision OMX V3 Blaze system (GE Healthcare). Images were analyzed as described previously[26], with all scripts and details of further script refinement to improve usability as detailed.

### ChrRNA-seq

ChrRNA was extracted according to a previous study[17]. Briefly, control or treated mES cells from one confluent 15-cm dish were trypsinized and washed in PBS. Cells were lysed on cold ice in RLB buffer (10 mM Tris pH 7.5, 10 mM KCl, 1.5 mM MgCl$_2$ and 0.1% NP-40) and nuclei were purified by centrifugation through a sucrose cushion (24% sucrose in RLB). Pelleted nuclei were resuspended in NUN1 (20 mM Tris-HCl pH 7.5, 75 mM NaCl, 0.5 mM EDTA and 50% glycerol) and then lysed with NUN2 (20 mM HEPES pH 7.9, 300 mM, 7.5 mM MgCl$_2$, 0.2 mM EDTA and 1 M urea). Samples were incubated for 15 min on ice with occasional shaking and then centrifuged at 2,800g to isolate the insoluble chromatin fraction. The chromatin pellet was resuspended in TRIzol by passing through a 23G needle several times. Finally, ChrRNA was purified through standard TRIzol–chloroform extraction followed by isopropanol precipitation. Samples were treated with two rounds of Turbo DNaseI to remove the DNA contamination. The quality of the ChrRNAs were checked by RNA bioanalyzer. Approximately 250–750 ng

of RNA was used for library preparation using the Illumina TruSeq stranded total RNA kit including the ribosomal RNA (rRNA) depletion step (RS-122-2301). Alternatively, Ribo-Magoff (Vazyme, N420) was used for rRNA removal. Libraries were quantified by qPCR with KAPA library quantification DNA standards (Kapa biosystems, KK4903). DNA and RNA concentrations were also determined by Qubit. The libraries were pooled and 2× 81-bp paired-end sequencing was performed using Illumina NextSeq 500 (FC-404-2002).

## ChrRNA-seq data analysis

The ChrRNA-seq data mapping pipeline used in this study was similar to previous work[17]. Briefly, the raw FASTQ files of read pairs were first mapped to an rRNA build by bowtie2 (version 2.3.5 or 2.4.5)[45] and rRNA-mapped reads were discarded. The remaining unmapped reads were aligned to the 'N-masked' genome (mm10) with STAR (version 2.5.2b or 2.7.9a)[46] using parameters '--twopassMode Basic --outSAMstrandField intronMotif --outFilterMismatchNoverReadLmax 0.06 --outFilterMultimapNmax 1 --alignEndsType EndToEnd' for all the sequencing libraries. Unique alignments were retained for further analysis. We made use of 23,005,850 SNPs between Cast and 129S genomes and used SNPsplit (version 0.4.0dev; Babraham Institute) to split the alignment into distinct alleles (Cast and 129S) using the parameter '--paired'. The (allelic) read numbers were counted by the program featureCounts (version 1.5.2) ('-t transcript -g gene_id -s 2')[47] and the alignments were sorted by SAMtools (version 1.16.1)[48]. Files of bigWig were generated by BEDTools (version 2.27.1)[49] and visualized by Integrated Genomics Viewer (IGV; version 2.17.1)[50] or UCSC Genome Browser. The metagene profile and heat map were generated by deepTools (version 3.5.5)[51] and custom Python scripts. For biallelic analysis, counts were normalized to 1 million mapped read pairs using the R package (version 4.1.0 or 4.2.1) edgeR. Genes with at least ten SNP-covering reads across all the samples were used to calculate the allelic ratio of Xi/(Xi + Xa). PCA was performed using the 'prcomp' function in R and plotted using the tidyverse (version 2.0.0) ggplot2 package. The allelic ratios of samples derived from SPEN knockout or deletion of Xist B/C-repeat were taken from the previous study[17]. Gene categories including initial X-linked gene expression level and promoter chromatin landscape were taken from previous studies[15,17] and gene silencing kinetics data were taken from another study[22].

## MeRIP-seq

MeRIP-seq was performed according to a previously described method[52] with minor modifications. Briefly, total RNA was isolated from preplated mES cells according to the procedure above. RNA was fragmented by incubation for 6 min at 94 °C in thin-walled PCR tubes with fragmentation buffer (100 mM Tris-HCl and 100 mM $ZnCl_2$). Fragmentation was quenched using stop buffer (200 mM EDTA pH 8.0) and incubation on ice more than 1 min, before ensuring the correct size (~100 nt) using RNA Bioanalyzer. Approximately 300 μg of fragmented (~100 nt) RNA was incubated with 10 μg of anti-$m^6A$ antibody (Synaptic Systems, 202 003), RNasin (Promega), 2 mM VRC, 50 mM Tris, 750 mM NaCl and 5% IGEPAL CA-630 in DNA/RNA low-bind tubes for 2 h before $m^6A$-containing RNA was isolated using 200 μl of protein A magnetic beads per immunoprecipitate (preblocked with BSA). After this 2-h incubation, extensive washing with 1× immunoprecipitation buffer (10 mM Tris-HCl pH 7.4, 150 mM NaCl and 0.1% NP-40), 2× low-salt buffer (50 mM Tris-HCl pH 7.4, 50 mM NaCl, 1 mM EDTA, 1% NP-40 and 0.1% SDS), 2× high-salt buffer (50 mM Tris-HCl pH 7.4, 1 M NaCl, 1 mM EDTA, 1% NP-40 and 0.1% SDS and 1× immunoprecipitation buffer was performed to remove the unspecific binding. Next, 6.7 mM $m^6A$ (Sigma-Aldrich) was used to elute RNA from the beads. Input and eluate samples were coprecipitated with ethanol and Glycoblue, quantified and pooled as libraries generated using TruSeq Stranded total RNA LT sample prep according to the manufacturer's instructions but skipping the fragmentation step. Finally, 75-bp single-end reads were obtained using Illumina NextSeq 500.

## MeRIP-seq data analysis

MeRIP-seq data analysis procedure was similar to ChrRNA-seq regarding the raw FASTQ read mapping. After mapping, sequencing reads were split into positive and negative strands using SAMtools (version 1.16.1)[48] and bigWig files were generated accordingly. A total of 10 million mapped reads per library were used to perform normalization. The previously defined $m^6A$ peak regions[21] including those on Xist RNA were taken for $m^6A$ intensity analysis, which is calculated as $log_2$(immunoprecipitate/input) with peak intensity close to 0 or less than 0 indicating no $m^6A$ enrichment. The selected peaks are visualized by IGV (version 2.17.1)[50].

## Total RNA-seq

METTL3 FKBP12$^{F36V}$-tagged cells (C3 and H5) were either treated with dTAG-13 for 26 h or not. Cells were washed with PBS twice and directly lysed with TRIzol, followed by total RNA isolation as described for ChrRNA-seq. DNA contamination was removed by Turbo DNase I treatment. The quality of the RNAs was checked by RNA bioanalyzer. Approximately 500 ng of total RNA was used for library preparation using the Illumina TruSeq stranded total RNA kit including the rRNA depletion step (RS-122-2301). The libraries were pooled and 2× 81-bp paired-end sequencing was performed using Illumina NextSeq 500 (FC-404-2002).

## Total RNA-seq data analysis

The total RNA-seq data analysis procedure was similar to that for ChrRNA-seq regarding the raw FASTQ reads mapping. PCR duplicates were removed using the Picard (version 2.25.0) command MarkDuplicates. Differentially expressed genes and transposable elements were called by TEtranscripts (version 2.2.1)[53] with default parameters. Upregulated and downregulated genes called were further cross-compared with $m^6A$ annotations from previous studies[21,54].

## SLAM-seq

SLAM-seq was performed using the SLAM-seq kinetics kit catabolic kinetics module (Lexogen, 062.24) as previously described[26] and shown in Extended Data Fig. 6a. Specifically, cells were grown in gelatin-coated six-well plates after preplating to discard feeder cells. When reaching 60–70% confluency, cells were either untreated or treated with dTAG-13 for 2 h, followed by additional 1 μg ml$^{-1}$ Dox induction for 20 h. Transcribed RNA was then labeled with 4sU by incubation with medium containing 500 μM 4sU (Lexogen) and 1 μg ml$^{-1}$ Dox for another 4 h. The 4sU was withdrawn for all samples by medium washout. Different samples were washed with medium containing 1 μg ml$^{-1}$ Dox and 50 mM uridine (in excess, Lexogen) for 1.5 and 3 h. Cells were washed with PBS once and immediately lysed with TRIzol–chloroform for total RNA isolation. Equal amounts of RNA (5 μg) were treated with iodoacetamide to modify the 4-thiol group of S4U-containing nucleotides through the addition of a carboxyamidomethyl group by the SLAM-seq Kinetics Kit (Lexogen). The RNA was recovered using RNAClean XP beads (Beckman Coulter), followed by resuspension in nuclease-free water. Approximately 500 ng of RNA from each sample was taken forward for library preparation using the Illumina TruSeq stranded total RNA kit (RS-122-2301). Quantification of the libraries was conducted by qPCR using KAPA library quantification DNA standards (Kapa Biosystems, KK4903). Finally, the libraries were pooled and 2× 81-bp paired-end sequencing was performed using Illumina NextSeq500 (FC-404-2002).

## SLAM-seq data analysis

Estimation of RNA half-life was performed using GRAND-SLAM[55]. Briefly, the paired-end sequencing reads were first aligned to rRNA

and then the unmapped reads were mapped to mouse genome mm10 by STAR (version 2.5.2b)[46] with the key parameters '--twopassMode Basic --outSAMstrandField intronMotif --outSAMattributes All --outFilterMultimapNmax 1 --outFilterMismatchNoverReadLmax 0.06 --alignEndsType EndToEnd'. Given that T-to-C conversion is the signature of SLAM-seq, the T-to-C conversion rate was calculated as 4sU incorporation and the average of all remaining conversions was calculated as background because of errors from sequencing or library preparation. The T-to-C conversion rate and background rate were calculated accordingly for each sample. The RNA decay was assumed to follow an exponential model; thus, the corresponding background corrected T-to-C conversion rates were fitted to the exponential model to estimate Xist RNA half-life.

### Reporting summary
Further information on research design is available in the Nature Portfolio Reporting Summary linked to this article.

### Data availability
High-throughput raw sequencing data and key processed data, including ChrRNA-seq, SLAM-seq, MeRIP-seq and total RNA-seq, were deposited to the National Center for Biotechnology Information's Gene Expression Omnibus (GEO) under accession number GSE279269. The mouse genome (mm10) sequence and gene annotation were downloaded from UCSC genome browser (https://hgdownload.soe.ucsc.edu/downloads.html). The whole-genome collections of SNPs and short indel variants for mouse strains 129S1 and Cast/EiJ (mpg.v5) were downloaded from the mouse genome project (https://www.sanger.ac.uk/data/mouse-genomes-project/). Gene categories including initial X-linked gene expression level were taken from the GEO under accession number GSE119602. Gene silencing kinetics data were taken from the GEO under accession number GSE185843. The promoter chromatin landscape of mm10 genome was retrieved from GitHub (https://github.com/guifengwei/ChromHMM_mESC_mm10). Source data are provided with this paper.

### Code availability
Key parameters used for sequencing data analysis are detailed in the Methods. The RNA-SPLIT analysis code was deposited to GitHub (https://github.com/HollyRoach/Automated_RNA-SPLIT).

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

### Acknowledgements
We are grateful to F. Constantinescu for assistance with mES cell tissue culture during pandemic, to J. Bowness for helping with the initial RNA-seq library preparation, to A. Cawte for helping with construct design for knock-in at the *Rosa26* locus, to Y. Zhang for quantifying site-specific m6A changes on Xist and to all the N.B. lab members for fruitful discussions. Imaging was carried out at the Micron Oxford Advanced Bioimaging Unit, Department of Biochemistry, University of Oxford. We thank Oxford Biochemistry IT support for computing server maintenance. We thank B. Moindrot for critical reading of the manuscript. This work is supported by Wellcome Trust (215513/Z/19/Z to N.B.) and UKRI (EP/Y029062/1 to N.B.). The funders had no role in study design, data collection and analysis, decision to publish or preparation of the manuscript.

### Author contributions
G.W. and N.B. conceptualized the study. G.W., with assistance from M.A., L.R. and T.B.N., generated the dTAG degron cell lines. G.W. conducted the ChrRNA-seq, SLAM-seq, total RNA-seq and MeRIP-seq and performed all sequencing data analyses. L.R., under the supervision of H.C. and N.B., developed RNA-SPLIT and applied it to Xist in this study. H.C., H.L.R. and L.R. analyzed the RNA-SPLIT data and produced the plots. N.B. and G.W. drafted the manuscript with contributions from coauthors. All authors read and approved the manuscript. N.B. supervised the entire project.

### Competing interests
The authors declare no competing interests.

### Additional information
**Extended data** is available for this paper at https://doi.org/10.1038/s41594-025-01663-w.

**Correspondence and requests for materials** should be addressed to Guifeng Wei or Neil Brockdorff.

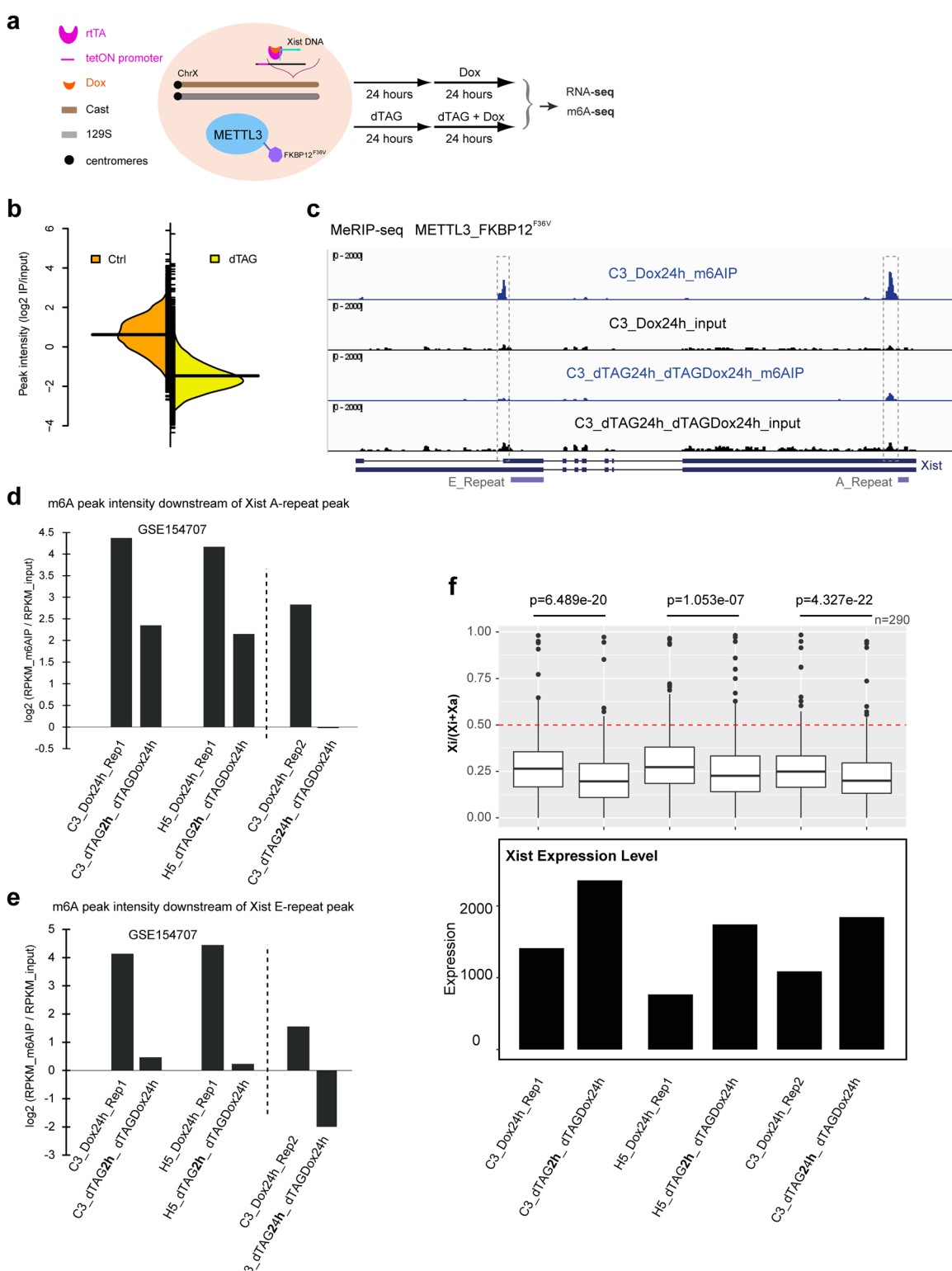

**Extended Data Fig. 1 | MeRIP confirms that m⁶A on Xist is dependent on METTL3 and acute depletion of METTL3 upregulates Xist and accelerates X-linked gene silencing. a**, Schematic illustrates the setup for the MeRIP experiment. **b**, Beanplot showing the m⁶A intensity distributions for previously defined m⁶A peaks. Orange and yellow back-to-back plots represent control (Ctrl) and dTAG conditions, respectively. The black solid lines denote the mean of each distribution. **c**, Genome Bowser view of MeRIP (blue) and input (black) signal across the Xist locus. The scale is indicated on the left, and Xist gene structure is shown below. The m⁶A peaks/clusters downstream of Xist A-repeat and downstream of Xist E-repeat are highlighted by dashed grey boxes. **d-e**, Barplots depicting m⁶A intensity for the representative peak located downstream of Xist

A-repeat (**d**) and downstream of Xist E-repeat (**e**). Sample names and conditions are indicated below, and y-axis represents m⁶A intensity, that is log2 transformed value. The first four samples are from our previous study (GSE154707) and last two samples are generated from this study. Each bar represents one sample. **f**, Top: boxplots showing the calculated allelic ratio from input samples that were used in MeRIP experiments, with the number of gene for this analysis indicated above. Centre lines indicate the median, box limits indicate the first and third quartiles, and whiskers indicate 1.5× IQR. P values were calculated by a two-sided paired t-test. Bottom, barplots showing the Xist RNA expression (RPM) corresponding to the top panel. Sample order is as in **d**. Each bar represents one sample.

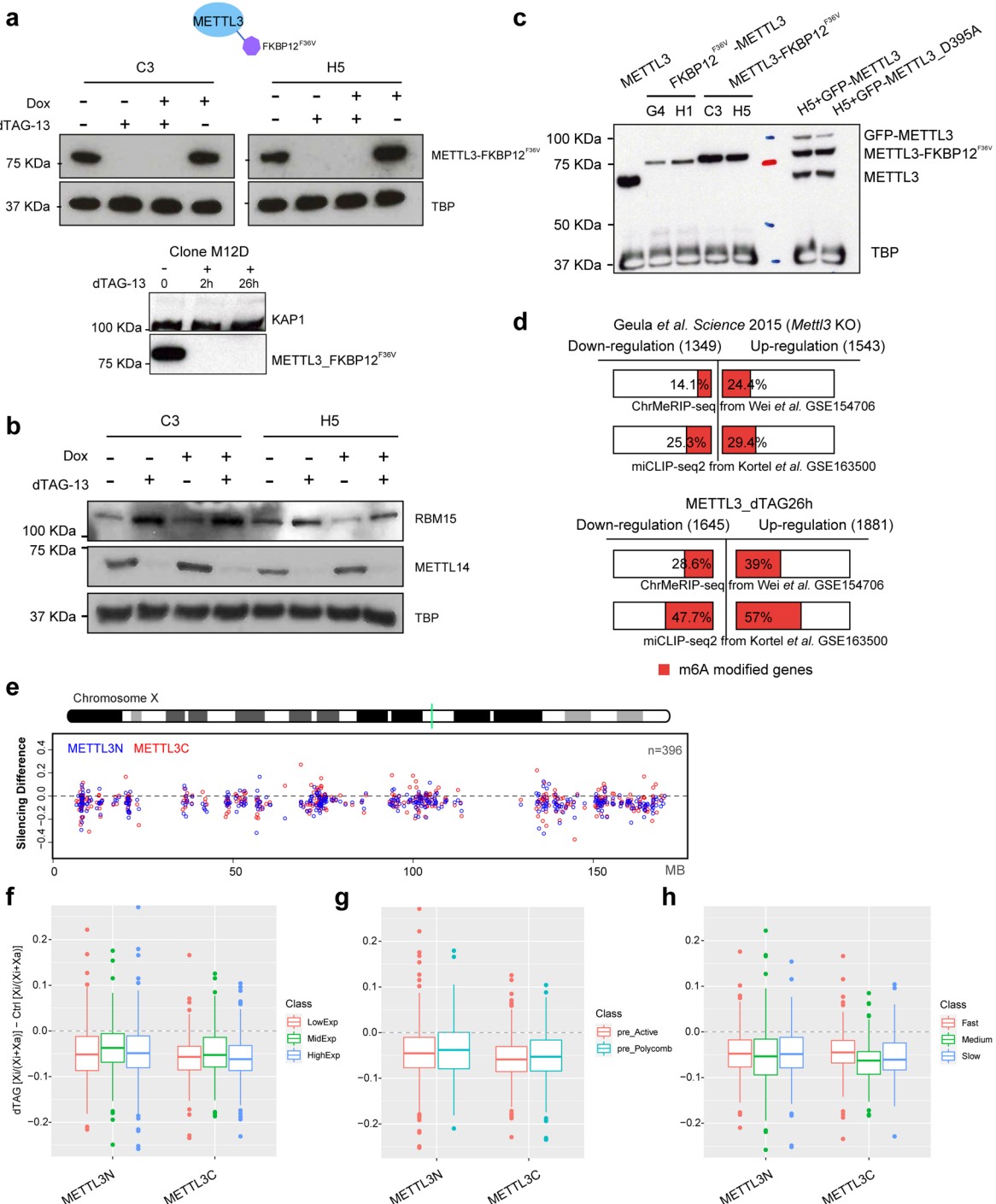

**Extended Data Fig. 2 | Derivation of degron tagged METTL3 cells and their effect on Xist-mediated silencing. a**, Western blot showing the METTL3-FKBP12^F36V protein level in two independent clones (C3 and H5) under ChrRNA-seq analysis conditions shown in Fig. 1d (top). TBP loading control. Western blot showing another clone of C-terminal METTL3 dTAG degron (M12D) generated in the Xist_Exon7-Bgl background exhibits sensitivity to dTAG-13 treatment at both 2 and 24 h (bottom). KAP1 loading control. **b**, Western blot showing the protein level for RBM15 and METTL14 for C3 and H5 clones under ChrRNA-seq analysis conditions shown in Fig. 1d. **c**, Western blot showing protein size and level for WT METTL3, FKBP12^F36V tagged METTL3, and GFP-tagged WT or catalytic mutant METTL3. TBP loading control. The blot is representative of two biologically independent experiments. **d**, Bar plot showing the proportion of differentially expressed genes in *Mettl3* KO mESCs[23] (top) or 26 h dTAG-13 treated METTL3-FKBP12^F36V mESCs (bottom) with

m^6A modification. The number of differentially up- or down-regulated genes are indicated. m^6A peaks/sites are annotated from ChrMeRIP-seq data[21] and miCLIP-seq2 data[54]. **e**, Allelic ratio difference between dTAG + Dox samples and Dox samples in Fig. 1d. Values from METTL3 N-terminal or C-terminal FKBP12^F36V are averaged and shown as METTL3N (blue) and METTL3C (red) respectively. Green line on ChrX ideogram indicates location of Xist. Dashed line denotes 0, indicating no silencing difference. **f-h**, As in (**e**), but for gene group analysis. Boxplots showing the difference of allelic ratio towards three different gene categories defined previously: initial X-linked gene expression level[17] (**f**), initial promoter chromatin state[15] (**g**), gene silencing kinetics[22] (**h**). FKBP12^F36V indicated as dTAG. Centre lines indicate the median, box limits indicate the first and third quartiles, and whiskers indicate 1.5× IQR.

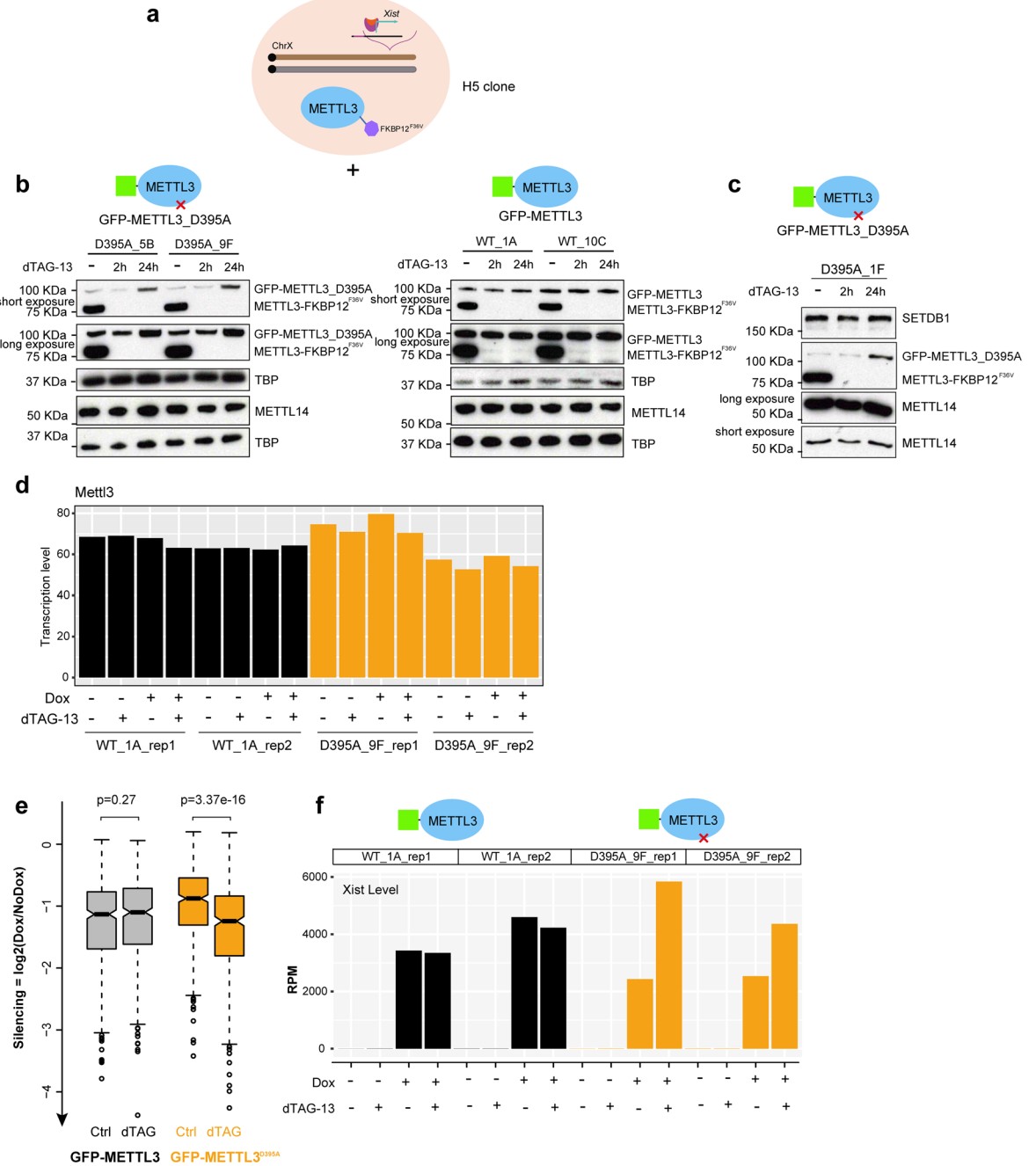

**Extended Data Fig. 3 | METTL3 catalytic activity is required for acceleration of Xist-mediated silencing. a**, Schematic showing the strategy for transgene complementation. METTL3 transgene is inserted into *Rosa26* locus and transcribed under the native *Rosa26* promoter. **b**, Western blots showing the protein level for GFP-METTL3$^{D395A}$ (left: 5B and 9F are two independent clones) and GFP-METTL3 (right: 1A and 10C are two independent clones), along with METTL3-FKBP12$^{F36V}$ and METTL14, after 2 or 24 h dTAG-13 treatment. The top and second from top gels are short and long exposure respectively. TBP and SETDB1 loading controls. Note that clones 5B, 9F, 1A, and 10C are X$_{cast}$0 due to loss of X$_{129}$. **c**, Western blots showing the protein level for GFP-METTL3$^{D395A}$, along with METTL3-FKBP12$^{F36V}$ and METTL14, after 2 or 24 h dTAG-13 treatment. The bottom and second from bottom gels are short and long exposure respectively. SETDB1 loading controls. Note that clone 1F has retained both X$_{129}$ and X$_{cast}$. Here X$_{cast}$ is the Xi chromosome. **d**, Barplot showing Mettl3 transcript level based on ChrRNA-seq data for cells harbouring GFP-METTL3 (represented by clone 1A)

or GFP-METTL3D$^{395A}$ (represented by clone 9F). Two replicates for each clone are included. Expression level is the sum of reads from native *Mettl3* locus and transgenic *Rosa26* locus. The yellow bars represent the level from cells expressing the transgene encoding METTL3$^{D395A}$. Each bar represents one sample. **e**, Boxplot showing average gene expression levels for X-linked genes in control or dTAG-13 treated samples. The averages are calculated from two clones shown in (**b**). Yellow boxplots represent the level from cells expressing the transgene encoding METTL3$^{D395A}$. Significance was determined by use of a two-tailed Wilcoxon test. Centre lines indicate the median, box limits indicate the first and third quartiles, and whiskers indicate 1.5× IQR. **f**, Xist expression level from samples of GFP-METTL3 or GFP-METTL3$^{D395A}$ clones. Clone names are indicated as above. Y-axis denotes the RPM value, as shown in Fig. 2a. Yellow barplots represent the level from cells expressing the transgene encoding METTL3$^{D395A}$. Each bar represents one sample.

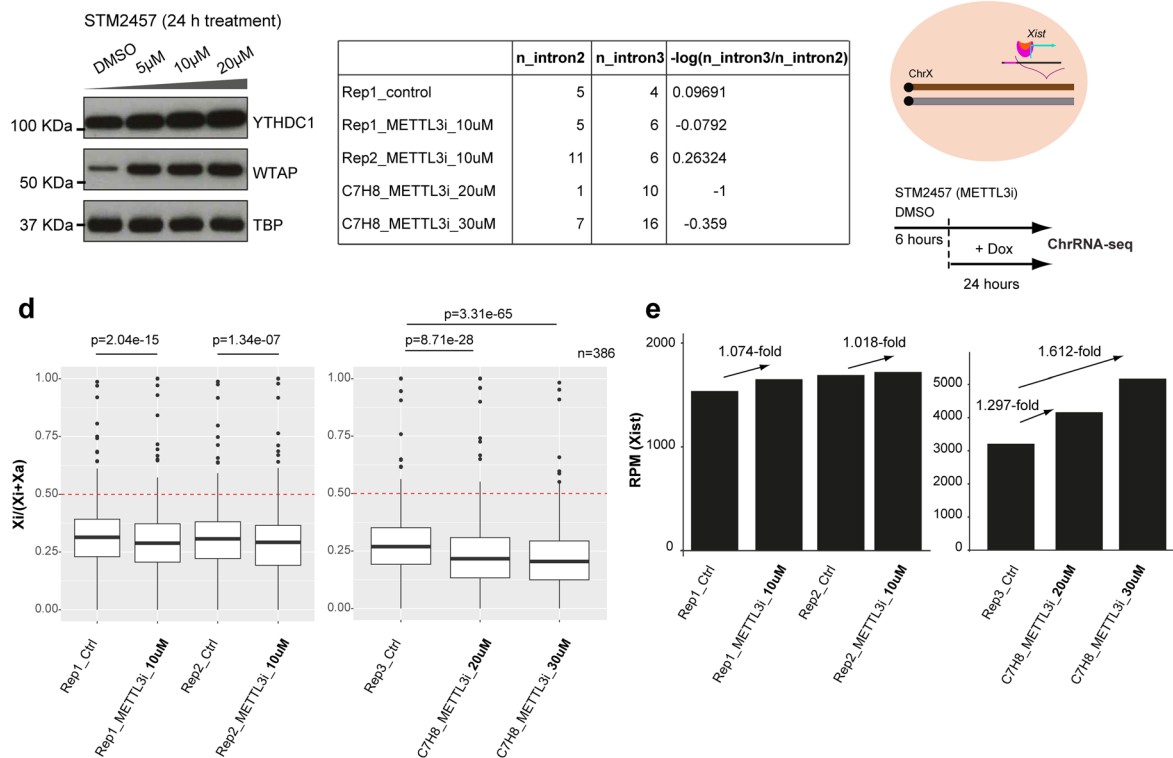

**Extended Data Fig. 4 | Xist upregulation and accelerated XCI dynamics upon METTL3 inhibition by STM2457. a**, Western blot showing increased YTHDC1 and WTAP protein levels following METTL3 inhibition with STM2457 (5, 10, and 20 μM). The blot is representative of three biologically independent experiments. **b**, Table summarizing the number of reads spanning intron 2 and intron 3 of *Tor1aip2* in control and STM2457-treated samples. **c**, Schematic of cell line and experimental design. **d**, Boxplot illustrating allelic ratios of X-linked genes from ChrRNA-seq analysis. Red lines indicate an allelic ratio of 0.5. P-values are calculated using a two-sided paired *t*-test. Centre lines indicate the median, box limits indicate the first and third quartiles, and whiskers indicate 1.5× IQR. **e**, Bar plot showing Xist expression levels (RPM, reads per million mapped reads) from ChrRNA-seq analysis across conditions described in (**c,d**). Fold changes are indicated. Each bar represents one biological replicate.

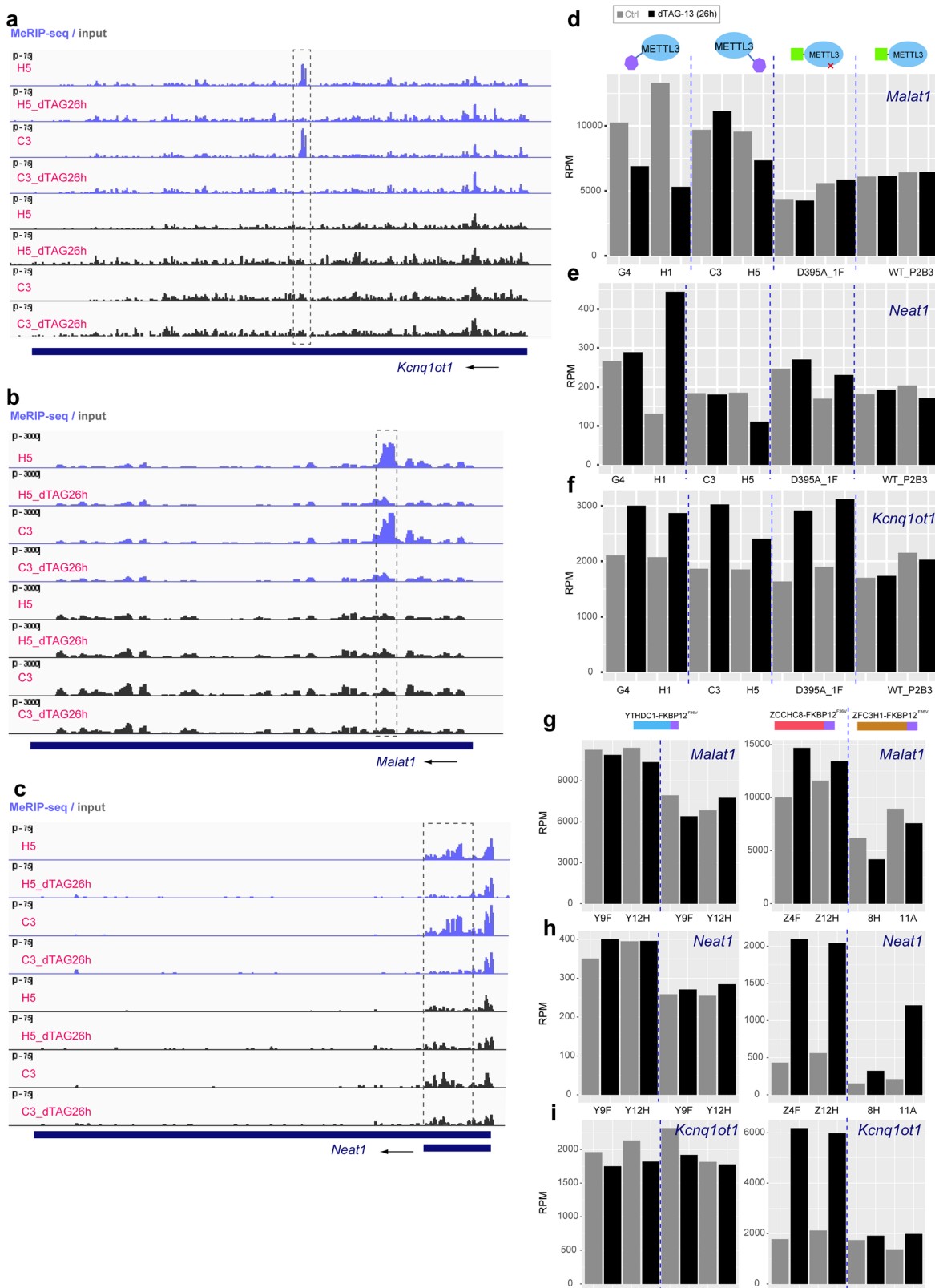

**Extended Data Fig. 5 | m⁶A landscape and gene expression analysis for lncRNAs Malat1, Neat1, and Kcnq1ot1. a-c**, m⁶A profile on lncRNAs Kcnq1ot1 (**a**), Malat1 (**b**), and Neat1 (**c**). Data are from ref. 21. MeRIP-seq data is shown in blue and input data in grey. For each panel, gene structure and transcriptional direction are shown below. Black boxes highlight METTL3 dependent m⁶A modification peaks. **d-f**, Expression levels for lncRNAs Malat1 (**d**), Neat1 (**e**), and Kcnq1ot1 (**f**) from ChrRNA-seq analysis. Cell lines are indicated at the top and clone names are indicated at the bottom. Samples are from FKBP12^F36V-METTL3, METTL3-

FKBP12^F36V and transgene complementation experiments. Control samples are shown in grey, whereas the dTAG-13 treated (26 h) samples are shown in black. **g-i**, Expression levels for lncRNAs Malat1 (**g**), Neat1 (**h**), and Kcnq1ot1 (**i**) from ChrRNA-seq analysis. Cell lines are indicated at the top and clone names are indicated at the bottom. Samples are from YTHDC1-FKBP12^F36V (left), ZCCHC8-FKBP12^F36V and ZFC3H1-FKBP12^F36V (right). Control samples are shown in grey, whereas the dTAG-13 treated (26 h) samples are shown in black. Note that two replicates are shown for each YTHDC1 clone.

**a**

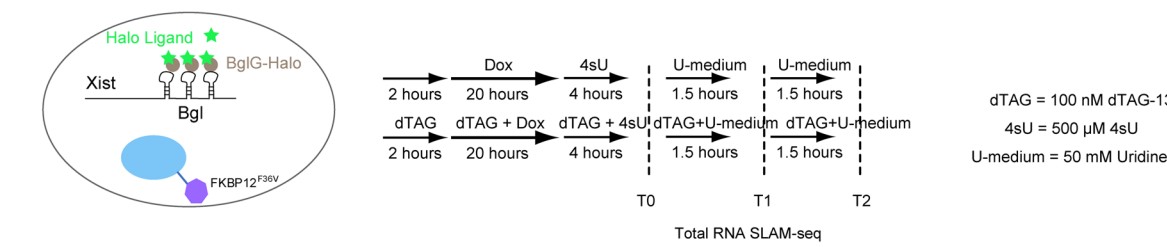

**b**

METTL3 dTAG degron
t1/2 = 180.8586 min
t1/2 = 184.031 min

YTHDC1 dTAG degron
t1/2 = 159.4043 min
t1/2 = 154.423 min

ZCCHC8 dTAG degron
t1/2 = 203.3467 min
t1/2 = 234.0293 min

METTL3 dTAG degron
t1/2 = 161.2502 min
t1/2 = 174.671 min

YTHDC1 dTAG degron
t1/2 = 136.9828 min
t1/2 = 135.6264 min

ZCCHC8 dTAG degron
t1/2 = 157.4782 min
t1/2 = 179.4043 min

**c**

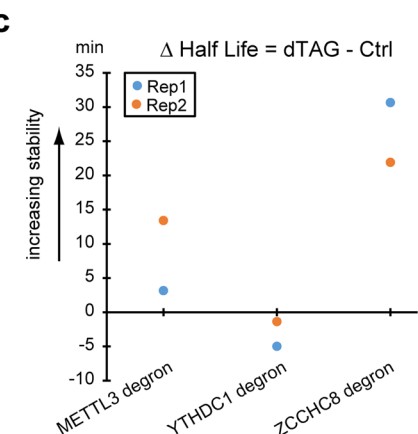

Extended Data Fig. 6 | See next page for caption.

**Extended Data Fig. 6 | Xist half-life estimated by SLAM-seq analysis.**
**a**, Schematic showing the experimental design for SLAM-seq. Cells used in this analysis are in the Xist-BglG expressing lines with -FKBP12$^{F36V}$ tags on METTL3, YTHDC1, or ZCCHC8. **b**, Best fit curves indicating Xist degradation determined from SLAM-seq T-to-C conversions. Two independent experiments are shown (top and bottom panels). Black and red represent untreated and dTAG-13 treated samples respectively. **c**, Dot plot showing the difference of Xist half-life with and without dTAG-13 treatment from (**b**). Replicates 1 and 2 are shown in blue and orange respectively.

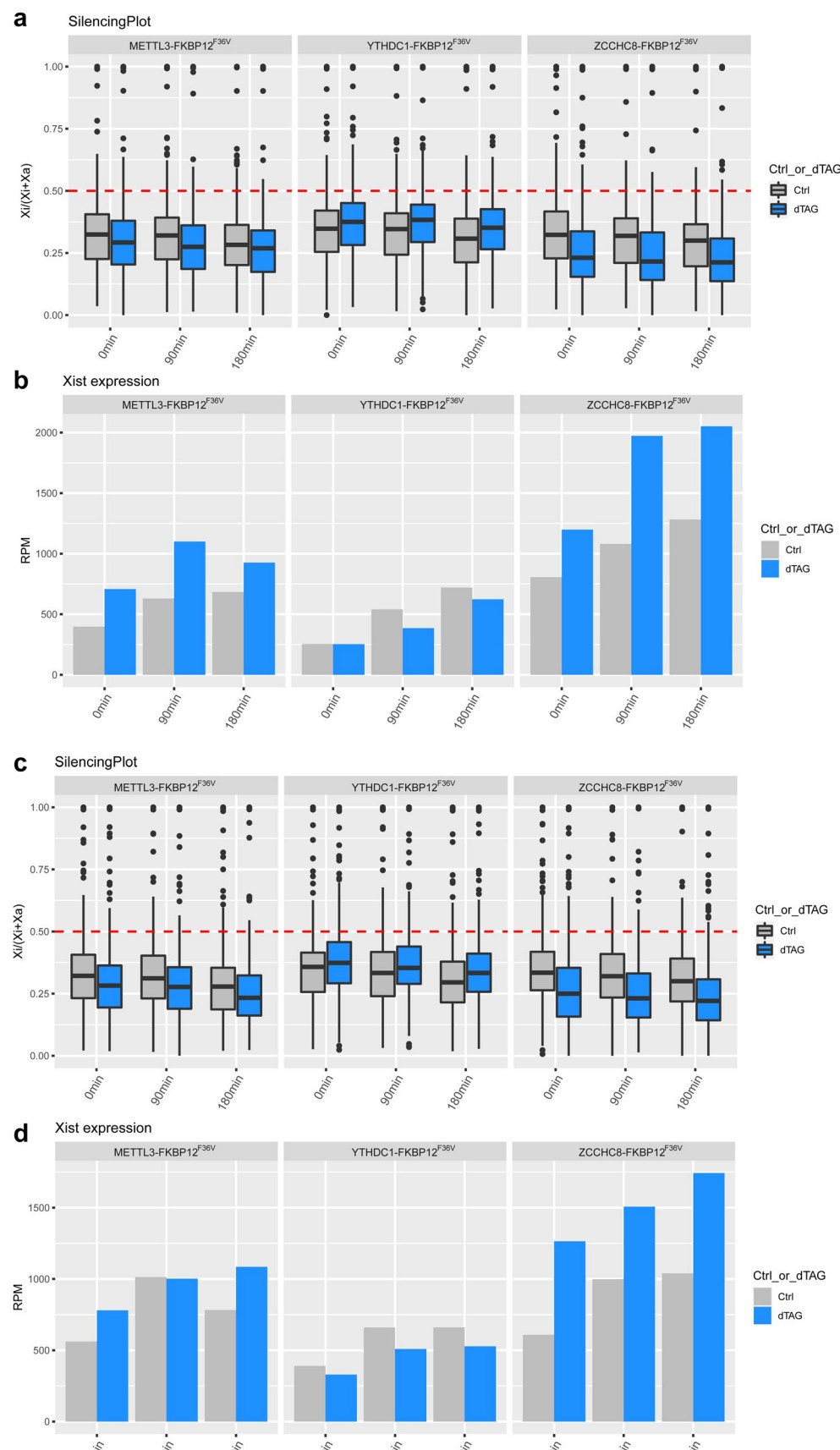

**Extended Data Fig. 7 | See next page for caption.**

**Extended Data Fig. 7 | X-inactivation dynamics and Xist levels calculated from SLAM-seq experiments. a**, Boxplot showing X-inactivation dynamics in the SLAM-seq timecourse experiment shown in Extended Data Fig. 6. Grey and blue indicate untreated and dTAG-13 treated samples respectively. The dotted line indicates an allelic ratio of 0.5. Centre lines indicate the median, box limits indicate the first and third quartiles, and whiskers indicate 1.5× IQR. **b**, Xist expression level in the samples shown in (**a**). **c**, Boxplot showing X-inactivation dynamics in the replicate 2 of SLAM-seq timecourse experiment, as in **a**. **d**, Xist expression level in the samples shown in (**c**).

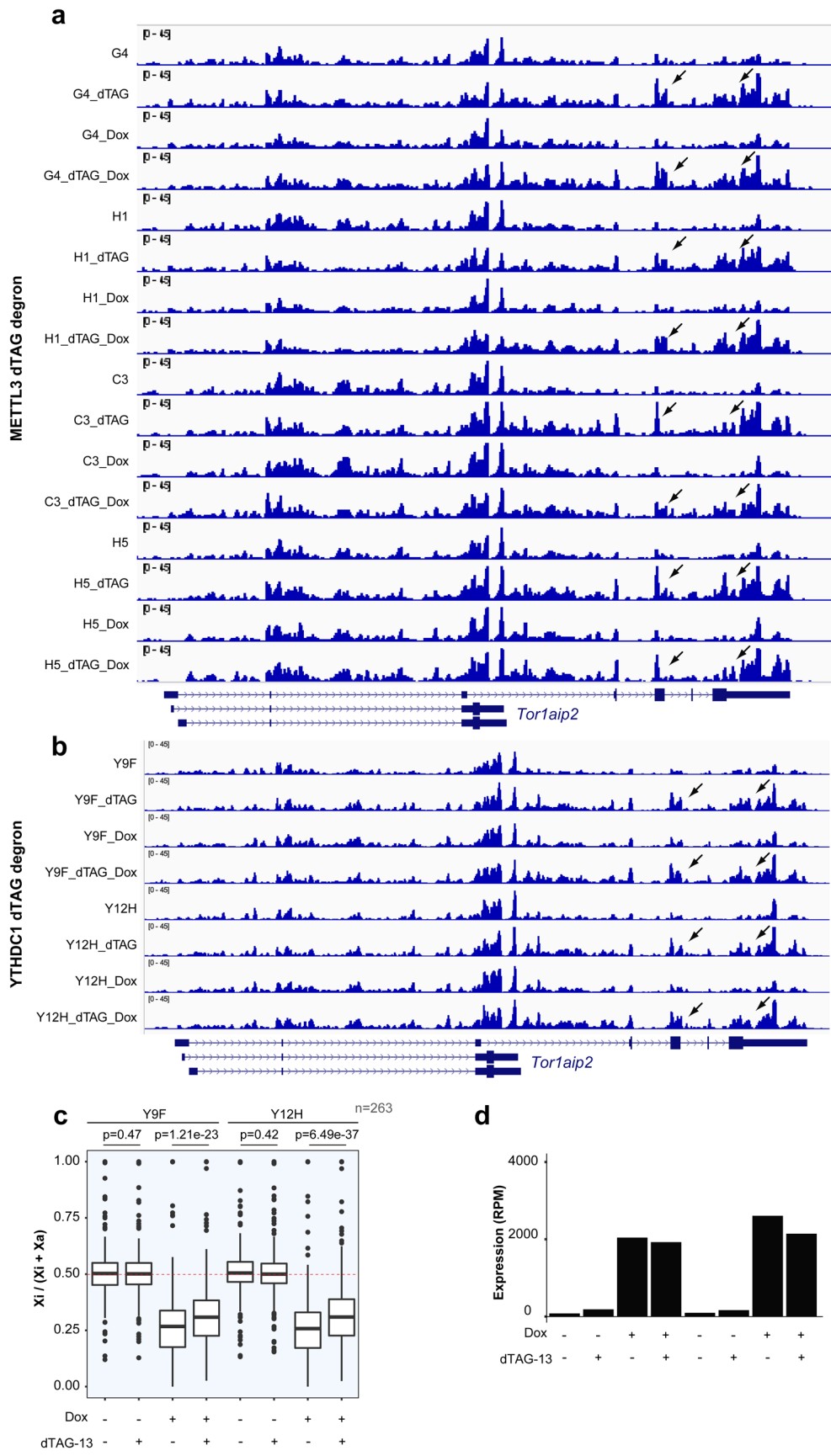

**Extended Data Fig. 8 | See next page for caption.**

**Extended Data Fig. 8 | FKBP12$^{F36V}$ degron for nuclear m$^6$A reader protein YTHDC1. a,b**, Profile of Tor1aip2 short and long isoforms in ChrRNA-seq samples of (**a**) METTL3-FKBP12$^{F36V}$ cells and (**b**) YTHDC1-FKBP12$^{F36V}$ degron cells (**b**). The arrows indicate the long isoform strongly upregulated in all of the dTAG-13 treated samples compared to untreated samples. Tor1aip2 gene structure is shown below. **c**, Boxplot showing the allelic ratio of X-linked genes (n = 263) from ChrRNA-seq analysis for YTHDC1 dTAG degron samples. The experimental design is as described in Fig. 1c. The red dashed line indicates allelic ratio at 0.5. Samples (two independent clones, Y9F and Y12H) and conditions are indicated above and below respectively. P-values, as indicated, are calculated using a two-sided paired *t*-test. Centre lines indicate the median, box limits indicate the first and third quartiles, and whiskers indicate 1.5× IQR. **d**, Barplot showing the expression level of Xist from ChrRNA-seq analysis for samples and conditions described in (**c**). Each bar represents one biological replicate.

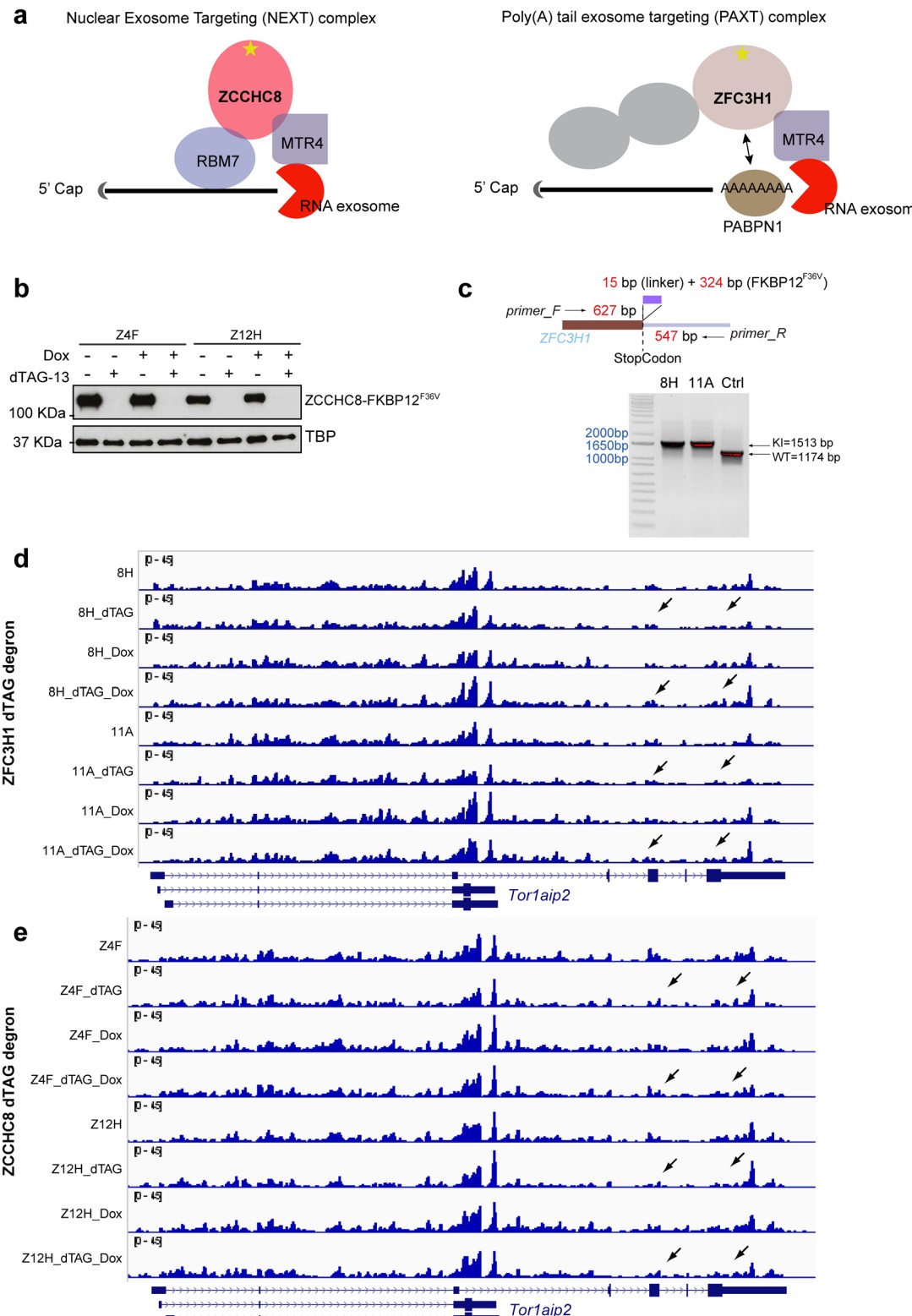

**Extended Data Fig. 9 | dTAG degron for ZFC3H1, a core component of PAXT complex, and ZCCHC8, a core component of NEXT complex. a**, Schematics illustrating the action of NEXT and PAXT complexes for nuclear RNA degradation. FKBP12$^{F36V}$ tagged subunit is indicated (star). **b**, Western blot verifying selected clones Z4F and Z12H in Fig. 5c. TBP loading control. **c**, Validation of FKBP12$^{F36V}$ insertion in the selected clones in Fig. 5b,d using PCR. **d**, Profile of Tor1aip2 short and long isoforms in ChrRNA-seq samples shown in Fig. 5d. The arrows indicate

that the long isoform is not upregulated in dTAG-13 treated samples compared to untreated samples. See also Extended Data Fig. 8 for comparison. Tor1aip2 gene structure is shown below. **e**, Profile of Tor1aip2 short and long isoforms in ChrRNA-seq samples shown in Fig. 5c. The arrows indicate that the long isoform is not upregulated in dTAG-13 treated samples compared to untreated samples. See also Extended Data Fig. 8 for comparison. Tor1aip2 gene structure is shown below.

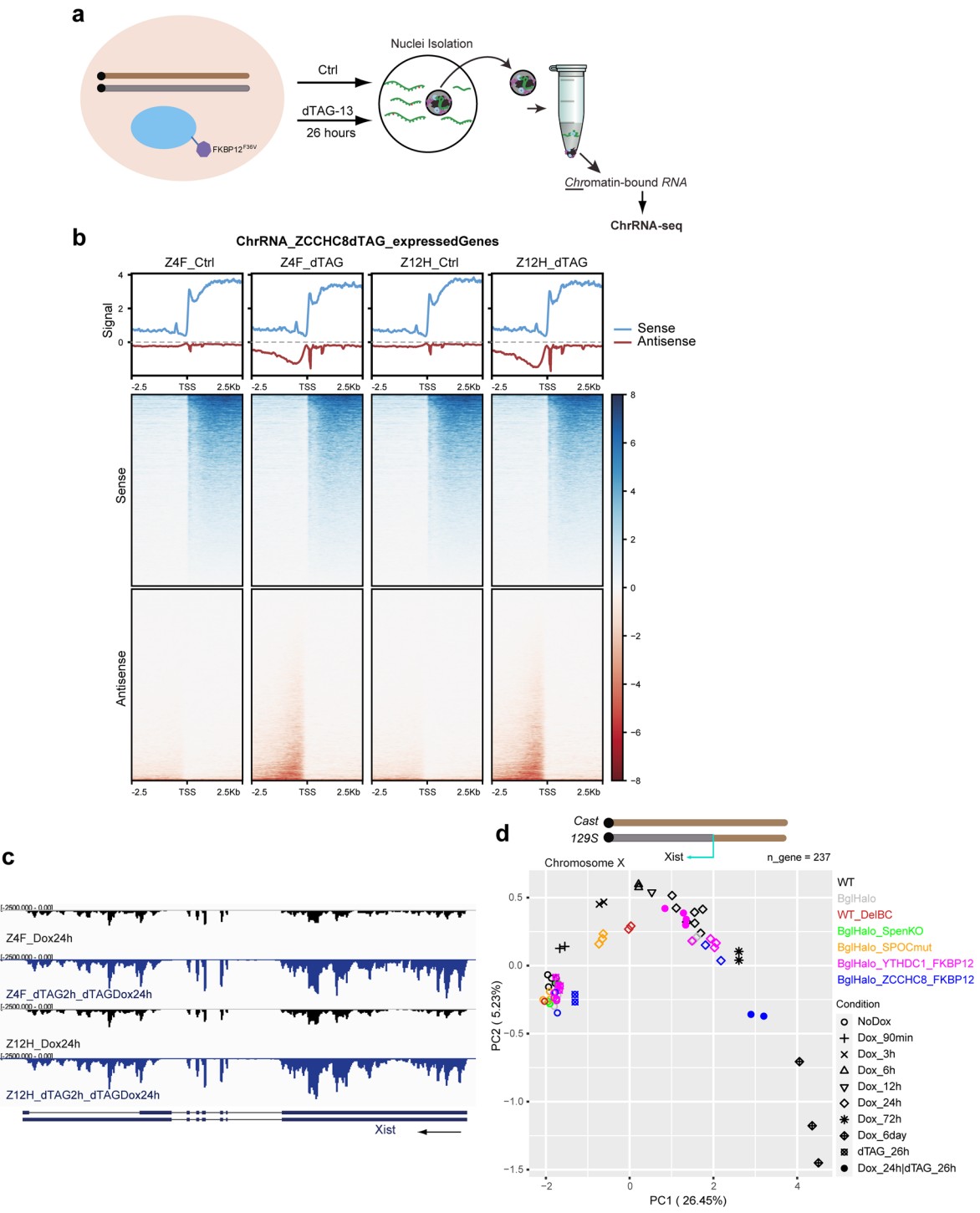

**Extended Data Fig. 10 | Expression profile of PROMPTs, Xist RNA, and X-linked silencing in ZCCHC8-FKBP12^F36V cells. a**, Schematic showing the experimental procedure for control or 26 h dTAG-13 treated samples for ChrRNA-seq analysis. **b**, Aggregated profile (top) and heatmap (bottom) showing the expression of expressed genes (blue) and their promoter upstream transcripts (PROMPTs) (red). The negative value shown in red in this plot indicates antisense transcription. Plots show data for transcription start site ± 2.5 kb. **c**, Genome browser screenshot of ChrRNA-seq of the Xist gene from ZCCHC8

degron samples with Dox or dTAG-13 + Dox treatment. **d**, PCA using allelic ratio of X-linked genes (n = 237) for samples of acute depletion of ZCCHC8 in Fig. 5 or YTHDC1 in Fig. 4, along with time-course WT (iXist-ChrX$_{129}$, A11B2 clone) cells, cells with SPEN knockout or SPOC mutant, and cells with Xist B/C-repeat deletion (data from ref. 17). Note that X-linked gene silencing is defective in SPEN knockout, SPOC mutant, and Xist B/C-repeat deletion, thus datapoints representing those samples lie away from the predicted silencing trajectory.

# Reporting Summary

## Statistics

For all statistical analyses, confirm that the following items are present in the figure legend, table legend, main text, or Methods section.

| n/a | Confirmed | |
|---|---|---|
| ☐ | ☒ | The exact sample size (*n*) for each experimental group/condition, given as a discrete number and unit of measurement |
| ☐ | ☒ | A statement on whether measurements were taken from distinct samples or whether the same sample was measured repeatedly |
| ☐ | ☒ | The statistical test(s) used AND whether they are one- or two-sided<br>*Only common tests should be described solely by name; describe more complex techniques in the Methods section.* |
| ☒ | ☐ | A description of all covariates tested |
| ☒ | ☐ | A description of any assumptions or corrections, such as tests of normality and adjustment for multiple comparisons |
| ☐ | ☒ | A full description of the statistical parameters including central tendency (e.g. means) or other basic estimates (e.g. regression coefficient) AND variation (e.g. standard deviation) or associated estimates of uncertainty (e.g. confidence intervals) |
| ☐ | ☒ | For null hypothesis testing, the test statistic (e.g. *F*, *t*, *r*) with confidence intervals, effect sizes, degrees of freedom and *P* value noted<br>*Give P values as exact values whenever suitable.* |
| ☒ | ☐ | For Bayesian analysis, information on the choice of priors and Markov chain Monte Carlo settings |
| ☒ | ☐ | For hierarchical and complex designs, identification of the appropriate level for tests and full reporting of outcomes |
| ☒ | ☐ | Estimates of effect sizes (e.g. Cohen's *d*, Pearson's *r*), indicating how they were calculated |

*Our web collection on statistics for biologists contains articles on many of the points above.*

## Software and code

Policy information about availability of computer code

| Data collection | Sequencing data for ChrRNA-seq, SLAM-seq, MeRIP-seq, and total RNA-seq were generated using the Illumina NextSeq 500 platform. RNA-SPLIT images were acquired with the DeltaVision OMX V3 Blaze system (GE Healthcare). Western blot images were developed using a Konica SRX-101A Medical Film Processor and scanned with a TASKalfa 5004i. qPCR was performed on the Rotor-Gene Q (QIAGEN). |
|---|---|
| Data analysis | Bowtie2 (2.3.5 & 2.4.5), SAMtools (1.16.1), STAR (v2.5.2b & 2.7.9a), SNPsplit (0.4.0dev), IGV (2.17.1), Subread (1.5.2), Picard tools (2.25.0), deeptools (3.5.5), bedtools (v2.27.1), Python (3.8.10), TEtranscripts (v2.2.1), R (4.1.0 & 4.2.1), and tidyverse (2.0.0). RNA-SPLIT analysis code has been deposited in github (https://github.com/HollyRoach/Automated_RNA-SPLIT). |

For manuscripts utilizing custom algorithms or software that are central to the research but not yet described in published literature, software must be made available to editors and reviewers. We strongly encourage code deposition in a community repository (e.g. GitHub). See the Nature Portfolio guidelines for submitting code & software for further information.

## Data

Policy information about availability of data

All manuscripts must include a data availability statement. This statement should provide the following information, where applicable:

- Accession codes, unique identifiers, or web links for publicly available datasets
- A description of any restrictions on data availability
- For clinical datasets or third party data, please ensure that the statement adheres to our policy

High-throughput raw sequencing data as well as key processed data, including ChrRNA-seq, SLAM-seq, MeRIP-seq, and total RNA-seq, are deposited to the National Center for Biotechnology Information's Gene Expression Omnibus (accession number GSE279269).
The mouse genome (mm10) sequence and gene annotation were downloaded from UCSC genome browser (https://hgdownload.soe.ucsc.edu/downloads.html). The whole genome collections of SNP and short indel variants for mouse strains 129S1 and Cast/EiJ (mpg.v5) was downloaded from mouse genome project (https://www.sanger.ac.uk/data/mouse-genomes-project/). Gene categories including initial X-linked gene expression level were taken from GSE119602. Gene silencing kinetics data were taken from GSE185843. Promoter chromatin landscape of mm10 genome were retrieved from (https://github.com/guifengwei/ChromHMM_mESC_mm10).
Uncropped western blots and numerical source data are available in source data. Previously published sequencing dataset and imaging dataset used in this study have been specified in the manuscript.

## Research involving human participants, their data, or biological material

Policy information about studies with human participants or human data. See also policy information about sex, gender (identity/presentation), and sexual orientation and race, ethnicity and racism.

| | |
|---|---|
| Reporting on sex and gender | NA |
| Reporting on race, ethnicity, or other socially relevant groupings | NA |
| Population characteristics | NA |
| Recruitment | NA |
| Ethics oversight | NA |

Note that full information on the approval of the study protocol must also be provided in the manuscript.

# Field-specific reporting

Please select the one below that is the best fit for your research. If you are not sure, read the appropriate sections before making your selection.

☒ Life sciences        ☐ Behavioural & social sciences        ☐ Ecological, evolutionary & environmental sciences

For a reference copy of the document with all sections, see nature.com/documents/nr-reporting-summary-flat.pdf

# Life sciences study design

All studies must disclose on these points even when the disclosure is negative.

| | |
|---|---|
| Sample size | No statistical methods were used to predetermine sample size for RNA-SPLIT and sequencing analysis.<br>For RNA-SPLIT, in theory, analyzing more cells leads to more accurate estimates. A minimum of 20 randomly selected cells at each time point was arbitrarily chosen for the analysis.<br>For western blots, at least two biologically independent replicates were performed.<br>For RNA-seq, either multiple independent clones or 2-3 independent repeats for a single clone were chosen, according to common practice in the field. This design ensures the results are reproducible. |
| Data exclusions | We confirm that no data were excluded for the analysis. |
| Replication | The number of biological replicates are indicated in the text, figure legend, or method section. |
| Randomization | For tissue culture based experiments, all wells in each biological replicate were split from the same batch of cells and randomly divided for each condition. |
| Blinding | For all the experiments and outcome assessments, the investigators were not blinded. This is because no subjective scoring was pre-required for this study as data analysis was all performed by software programs/algorithms. |

# Reporting for specific materials, systems and methods

We require information from authors about some types of materials, experimental systems and methods used in many studies. Here, indicate whether each material, system or method listed is relevant to your study. If you are not sure if a list item applies to your research, read the appropriate section before selecting a response.

## Materials & experimental systems

| n/a | Involved in the study |
|---|---|
| ☐ | ☒ Antibodies |
| ☐ | ☒ Eukaryotic cell lines |
| ☒ | ☐ Palaeontology and archaeology |
| ☒ | ☐ Animals and other organisms |
| ☒ | ☐ Clinical data |
| ☒ | ☐ Dual use research of concern |
| ☒ | ☐ Plants |

## Methods

| n/a | Involved in the study |
|---|---|
| ☒ | ☐ ChIP-seq |
| ☒ | ☐ Flow cytometry |
| ☒ | ☐ MRI-based neuroimaging |

## Antibodies

| Antibodies used | Primary antibody:<br>anti-METTL3 (Abcam, ab195352, 1:1000), anti-METTL14 (Sigma-Aldrich, HPA038002, 1:1000), anti-RBM15 (Proteintech, 10587-1-AP, 1:1000), anti-YTHDC1 (Sigma-Aldrich, HPA036462, 1:1000), anti-WTAP (Proteintech, 10200-1-AP, 1:1000), anti-m6A (Synaptic Systems, 202 003), anti-TBP (Abcam, ab51841, 1:1000), anti-SETDB1 (Proteintech, 11231-1-AP, 1:1000), anti-ZCCHC8 (Proteintech, 23374-1-AP, 1:1000), anti-ZFC3H1 (Sigma-Aldrich, HPA007151, 1:1000), and anti-KAP1 (Abcam, ab10484, 1:1000).<br><br>Secondary antibody:<br>anti-rabbit IgG HRP Donkey (Amersham, NA934V, 1:2000) and anti-mouse IgG HRP Sheep (Amersham, NXA931V, 1:2000). |
|---|---|
| Validation | Antibodies are verified by manufacturers using knockouts according to their websites.<br>Antibodies against mouse METTL3, YTHDC1, ZCCHC8, ZFC3H1 are also verified in this study because the FKBP-V insertion causes the upshift of the protein and the fusion proteins are sensitive to dTAG-13 treatment.<br>Antibodies against WTAP and RBM15 are also verified in this study by gene knockout.<br><br>anti-METTL3 (https://www.abcam.com/en-us/products/primary-antibodies/mettl3-antibody-epr18810-ab195352)<br>anti-METTL14 (https://www.sigmaaldrich.com/GB/en/product/sigma/hpa038002)<br>anti-RBM15 (https://www.ptglab.com/products/RBM15-Antibody-10587-1-AP.htm)<br>anti-YTHDC1 (https://www.sigmaaldrich.com/GB/en/product/sigma/hpa036462)<br>anti-WTAP (https://www.ptglab.com/products/WTAP-Antibody-10200-1-AP.htm)<br>Anti-m6A (https://sysy.com/product/202003)<br>anti-TBP (https://www.abcam.com/en-us/products/primary-antibodies/tata-binding-protein-tbp-antibody-mabcam51841-bsa-and-azide-free-ab282715)<br>anti-SETDB1 (https://www.ptglab.com/products/SETDB1-Antibody-11231-1-AP.htm)<br>anti-ZCCHC8 (https://www.ptglab.com/products/ZCCHC8-Antibody-23374-1-AP.htm)<br>anti-ZFC3H1 (https://www.sigmaaldrich.com/GB/en/product/sigma/hpa007151)<br>anti-KAP1 (https://www.abcam.com/en-us/products/primary-antibodies/kap1-antibody-ab10484)<br>anti-rabbit IgG HRP Donkey (https://www.cytivalifesciences.com/en/us/shop/protein-analysis/blotting-and-detection/blotting-standards-and-reagents/amersham-ecl-hrp-conjugated-antibodies-p-06260).<br>anti-mouse IgG HRP Sheep (https://www.cytivalifesciences.com/en/us/shop/protein-analysis/blotting-and-detection/blotting-standards-and-reagents/amersham-ecl-hrp-conjugated-antibodies-p-06260). |

## Eukaryotic cell lines

Policy information about cell lines and Sex and Gender in Research

| Cell line source(s) | All mouse embryonic stem cells (mESCs) used in this study are female, derived from the F1 2–1 XX mESC line (129/Sv-Cast/Ei), a gift from by J. Gribnau.<br><br>Using this background, we developed dox-inducible endogenous Xist cell lines targeted to either the 129S allele (iXist-ChrX129) or the Cast allele (iXist-ChrXCast). For the RNA-SPLIT experiments, we introduced a bgl stem-loop into Xist exon 7 (iXist-bgl) within the iXist-ChrX129 background.<br><br>We created N-terminal or C-terminal FKBP-V tagged METTL3 lines on the iXist-ChrXCast background, and C-terminal FKBP-V tagged METTL3 on iXist-bgl-ChrX129. We also generated C-terminal FKBP-V tagged YTHDC1 and ZCCHC8 were established in the iXist-bgl-ChrX129 line, and C-terminal FKBP-V tagged ZFC3H1 in iXist-ChrXCast. To further analyse METTL3 catalytic function, GFP-METTL3 and GFP-METTL3D395A lines were introduced as transgenes at the Rosa26 locus with expression driven by the Rosa26 promoter in one of the C-terminal FKBP-V tagged METTL3 lines in iXist-ChrXCast.<br><br>We generated Xist exon 7 m6A region deletions either alone or in combination with Xist exon 1 m6A region deletion in the |
|---|---|

iXist-ChrX129 background.

The karyotype of the X chromosome in each cell line was confirmed by PCR, as documented in the manuscript.

| Authentication | All engineered cell lines were validated at both the genomic level, using PCR, and at the protein level, using Western blot analysis. GFP-METTL3 transgene lines were further validated through microscopy to assess localisation. |
|---|---|
| Mycoplasma contamination | All the cell lines are regularly tested for mycoplasma contamination, and confirmed to be negative. |
| Commonly misidentified lines (See ICLAC register) | No commonly misidentified lines were used in this work. |

## Plants

| Seed stocks | Plants are not used in this study. |
|---|---|
| Novel plant genotypes | Plants are not used in this study. |
| Authentication | Plants are not used in this study. |

