## [Peer Review File · Nature Structural & Molecular Biology]

m6A and the NEXT complex direct Xist RNA turnover and X inactivation dynamics

Corresponding Author: Professor Neil Brockdorff

Version 0:

Decision Letter:

2nd Dec 2024

Dear Professor Brockdorff,

Thank you again for submitting your manuscript "N6-methyladenosine and the NEXT complex direct Xist RNA turnover and X inactivation dynamics". We now have comments (below) from the 3 reviewers who evaluated your paper. In light of those reports, we remain interested in your study and would like to see your response to the comments of the referees, in the form of a revised manuscript.

You will see that though the experts appreciate the potential functional novelty of the findings, they nevertheless raise important concerns that need to be convincingly addressed in a revised manuscript. More specifically, Reviewer #1 assesses that more Xist-specific approaches need to be employed to overrule potential Mettl3-depletion secondary effects. Along the same lines, both Reviewers #2 and #3 request that you perform Mettl3-inhibition experiments to validate the acute depletion phenotypes on Xist stability and they also ask that you investigate the effects of depleting WTAP and m6A readers. Importantly, Reviewer #1 notes the lack of important controls and of sufficient mechanistic insight, also pointing out the need to delineate which parts of Xist and which of its m6A sites are needed for the observed phenotypes. Please be sure to address/respond to all concerns of the referees in full in a point-by-point response and highlight all changes in the revised manuscript text file. If you have comments that are intended for editors only, please include those in a separate cover letter.

We expect to see your revised manuscript within 3-6 months. If you cannot send it within this time, please contact us to discuss an extension; we would still consider your revision, provided that no similar work has been accepted for publication at NSMB or published elsewhere.

Reporting Summary:

When submitting the revised version of your manuscript, please pay close attention to our <https://www.nature.com/nature-portfolio/editorial-policies/image-integrity> Digital Image Integrity Guidelines and to the following points below:

Finally, please ensure that you retain unprocessed data and metadata files after publication, ideally archiving data in

perpetuity, as these may be requested during the peer review and production process or after publication if any issues arise.

Data availability: this journal strongly supports public availability of data. All data used in accepted papers should be available via a public data repository, or alternatively, as Supplementary Information. If data can only be shared on request, please explain why in your Data Availability Statement, and also in the correspondence with your editor. Please note that for some data types, deposition in a public repository is mandatory - more information on our data deposition policies and available repositories can be found below:

<https://www.nature.com/nature-research/editorial-policies/reporting-standards#availability-of-data>

Link Redacted

Sincerely,

Dimitris Typas
Senior Editor
Nature Structural & Molecular Biology
ORCID: 0000-0002-8737-1319

Reviewers' Comments:

Reviewer #1 (Remarks to the Author):

Summary

In this manuscript, the authors use an acute depletion method to eliminate the m6A methyltransferase and conduct a series of experiments to characterize the behavior of the Xist long noncoding RNA involved in mammalian dosage compensation. They find that acute depletion m6A increases the overall levels of Xist, promoting better X-linked gene repression. Depletion of the primary nuclear m6A reader, YTHDC1, did not phenocopy depletion of m6A. Increased Xist stability was potentially due to the disruption of the NEXT/exosome mechanism of RNA degradation.

The study uses a powerful acute depletion approach to better assess the direct effects of eliminating m6A on Xist, though this still causes all m6A to be depleted, across the transcriptome. Any potential player in this pathway that is directly and acutely influenced by loss of m6A could be involved in this mechanism. To address this major limitation of the study, the authors would benefit by more Xist-specific experiments to disrupt m6A, as well as taking advantage of previous studies

from the PI that have already genetically modified regions of m6A methylation on Xist to put the current work in better context. The study is also severely limited by a lack of statistical analysis throughout the manuscript, with few exceptions. Finally, although certain pathways are ruled in and ruled out as candidates to regulate the mechanism of Xist stability, the study does not achieve mechanistic insight into what may connect m6A and NEXT, if they indeed are connected.

The following are major points that would strengthen the manuscript, if addressed:

Major Points

1. The degron approach is powerful. However, it would benefit from a control that, when acutely depleted, actually causes a significant disruption of X-linked gene repression. SPEN may be the most-likely candidate to compare. As it stands, the slight effect of YTHDC1 depletion has no appropriate point of comparison to assess if it is or is not as important as other Xist-related repression machinery
2. It would be very helpful to show data that demonstrate the peaks of m6A in Xist and how they are affected by both METTL3 degrons in the context of multiple important experiments used in this study. Some of this is already published in Ref. 21. It would strengthen the manuscript to reproduce those and add to them here.
3. Statistical analyses are lacking in all RNA-Seq experiments beginning with Figure 1D, which limits the impact of the study.
4. Xist has two main regions that have been identified as m6A methylated on multiple adenosines, one near RepA and one Near Rep E. The Brockdorff group has previously made precise mutations in the 5' region in RepA, with little apparent effect on stability, in Ref. 18. It is surprising this was not described more clearly in the current manuscript, or used in any experiments to compare to the more extreme approach of full acute m6A depletion. This highlights another major limitation of this study in not being able to piece apart the detailed mechanism of which m6A region(s) on Xist may contribute to the stability mechanism described here. The authors should seriously consider some method of targeted disruption of the 3' m6A region as well. If not by genetics, then potentially by RNA-targeting Cas protein fused to FTO/ALKBH5. This would make a major contribution to strengthening the manuscript's impact, by identifying which part of Xist and which m6A sites may be important.
5. Fig. 2G: by eye, it appears there are equal, if not more, individual objects/centroids in WT vs. Degron in the Expansion images. It would be helpful to clarify why this is the case. Is this due to thresholding? The volume differences in the representative image also appear to not follow the trend from the cumulative plot. Degron does not look like a smaller volume.
6. Fig. 3C: a plot of total Xist would be informative, at least in the supplemental, to determine if, in this window, the absolute amount of Xist is compromised, or if the exchange of old for new is the primary point of disruption.
7. As a point of discussion, is it helpful to address the "coupling" phenomenon that the Brockdorff group described in Ref. 25.

Minor Points

- A simplification of the clone names, perhaps simple "Clone #1 and Clone #2" would help reduce the need to keep track of alphanumeric clone names.
- Figure 1d: is there an appropriate control that can be shown with dTAG-13 ligand in the C7H8 parental line, +/- dox? This would better match the rest of the figure. In Figures 1d and 2a the parental line has replicates with no dTAG-13 instead, which does not match the rest of the experiments well.
- A better introduction to the Expansion and Steady-State windows would help the reader appreciate these two distinct phases.
- Line 177: the "*" likely refers to just one of the two p-values. Either generalize or add specific values for each plot.
- Figure 2d can go in supplement.
- Line 285-287: the PC analysis could be better described here.
- Line 353-355: is this model described in reverse? Wouldn't m6A promote the association of RBPs that would promote NEXT/exosome activity?
- When discussing m6A effects on lncRNA stability, it may be helpful to reference studies that found m6A sites can promote nuclear lncRNA stability, as a point of comparison (PMID 34048709; PMID 36441764).

Reviewer #2 (Remarks to the Author):

The authors of this manuscript report a careful analysis of the roles of m6A on Xist function. This manuscript shows a destabilizing effect of m6A on Xist RNA. In contrast to a previous report acute METTL3 depletion (m6A depletion) led to increased Xist level and X inactivation activity independent of YTHDC1. The authors also showed that the NEXT complex (ZCCHC8) is involved in Xist turnover. The results are convincing and should be published. I do have several comments:

1. There is a very good METTL3 inhibitor developed by Tony Kouzarides and colleagues. I strongly suggest applying this inhibitor for acute inhibition of METTL3. This could resolve the concern of WT versus mutant rescue and convincingly show

the activity dependency of Xist RNA turnover regulated through m6A.

2. This one could be optional. There is a quite selective inhibitor of YTHDC1. See: <https://www.researchsquare.com/article/rs-2644364/v1>. This molecule works quite well in our hands in cell-based studies. It would be nice to synthesize this molecule and test acute inhibition of YTHDC1 in the current system.

3. It would be nice if the authors can KD a few selected m6A binding proteins to see if they are involved in Xist degradation. hnRCP, hnRNPA2B1, FXR are obvious candidates. I think it will be nice to test a few known candidates. If they do not work at least the authors can exclude them.

Reviewer #3 (Remarks to the Author):

The manuscript entitled "N6-methyladenosine and the NEXT complex direct Xist RNA turnover and X inactivation dynamics" by Wei et al. investigates the role of RNA methylation on Xist function and the initiation of X chromosome inactivation in a mouse embryonic stem cell model. The authors use degron mediated depletion of the catalytic subunit of the N6-adenine methyltransferase complex to study the requirement for the establishment of X-linked gene repression, which in the study is mediated by Doxycycline regulated Xist expression. A main finding of the study is that Xist initiates efficient gene repression after depletion, which corrects earlier conclusions on a requirement of N6-A RNA methylation for gene silencing. The primary effect of RNA methylation appears destabilization of Xist RNA and after depletion Xist levels are increased at early timepoints after induction which leads to slightly increased or faster gene silencing. The authors go on to investigate Xist turnover by two independent methods RNA split using 3D-SIM superresolution microscopy and SLAMseq. Both methods show stabilization of Xist after Mettl3 depletion. Mechanistically Mettl3 appears to trigger degradation via a pathway involving ZCCHC8 but requires neither ZFC3H1 nor YTHDC1. The study provides convincing genetic evidence using degron-tagged alleles of the respective genes. However, the link to RNA methylation remains to be fully understood in future studies.

The paper stands out for the large amount of sophisticated evidence and clear conclusions. It is important for correcting the role of RNA methylation for Xist function. Although, the authors have already voiced concerns over a requirement in gene silencing this study appears to fully solve the issue and is important for the field. A few points that need clarification should be considered by the authors to further improve the presentation of the study.

Specific points

1. The authors find that Xist becomes more stable after depletion of Mettl3. If Mettl3 indeed were a major pathway for Xist degradation one could expect that Xist accumulates in the nucleus and potentially localizes to other chromosomes. The authors argue that this is not observed because of a negative effect on Xist transcription. I did not fully understand how this effect should operate. Firstly, Xist expression is said to be controlled by a "TetON" promoter and this raises the suspicion that it might not be fully relevant for regulation of the Xist promoter. Secondly, references to a negative effect on Xist transcription are to the authors own previous work but do not identify a plausible mechanism. It would be important to clarify this aspect in the manuscript.

2. Moindrot et al. (REF 10) and Chu et al. (REF 8) also identified WTAP as an Xist associated protein, which links to RNA methylation. Chu et al. show that WTAP is recruited to Xist dependent on gene repression. It would be interesting to consider WTAP either as a potential link between RNA methylation and gene silencing or explain why this has not been considered in the study.

3. The authors show that METTL14 protein is strongly reduced after degron-mediated depletion of METTL3. This brings up the concern that the degron might affect not only the target but also associated proteins, which in the context of chromatin could potentially also include components of the transcription machinery. As the present study seems to contradict earlier conclusions of a requirement of METTL3 for gene silencing, it would be important to consider such effects. Would chemical inhibition of the METTL3/14 complex also lead to Xist stabilization and more rapid or effective silencing?

Minor points

a) Can cell cycle effects be ruled out to lead to an increase of Xist RNA after acute depletion of Mettl3? As Xist displacement from chromatin in M-phase contributes to turnover a statement could be added if cell cycle differences might have been observed.

b) I found it puzzling that RNA methylation but not YTHDC1 had an effect on Xist stabilization. It would be interesting to have the authors' opinion on what YTHDC1 function might be or indeed if it binds Xist.

c) line 285: after "5d" the closing parenthesis seems missing

Version 1:

Decision Letter:

Our ref: NSMB-A49964A

20th May 2025

Dear Professor Brockdorff,

Thank you for submitting your revised manuscript "N6-methyladenosine and the NEXT complex direct Xist RNA turnover and X inactivation dynamics" (NSMB-A49964A). It has now been seen by the original referees and their comments are below. The reviewers find that the paper has improved in revision, and therefore we can now accept it in principle in Nature Structural & Molecular Biology, pending minor revisions to satisfy the referees' final requests and to comply with our editorial and formatting guidelines.

We are now performing detailed checks on your paper and will send you a checklist detailing our editorial and formatting requirements in about two weeks. Please do not upload the final materials and make any revisions until you receive this additional information from us.

To facilitate our work at this stage, it is important that we have a copy of the main text as a word file. If you could please send along a word version of this file as soon as possible, we would greatly appreciate it; please make sure to copy the NSMB account (cc'ed above).

Sincerely,

Dimitris Typas
Senior Editor
Nature Structural & Molecular Biology
ORCID: 0000-0002-8737-1319

Reviewer #1 (Remarks to the Author):

This revised manuscript has met all of the suggestions that this reviewer deemed important for publication. Although this is at the author's discretion, this reviewer encourages the authors to at the very least include a mention of the exon 7 m6A site deletion result that was described only in the rebuttal. Even with the somewhat complex possibilities for interpretation, the result generally fits with the model and suggests a possible RNA stability sequence that is negatively regulated by m6A, which the authors note. This information is valuable to the field.

Reviewer #2 (Remarks to the Author):

The authors have addressed my comments.

Reviewer #3 (Remarks to the Author):

The revised version of the manuscript entitled "N6-methyladenosine and the NEXT complex direct Xist RNA turnover and X inactivation dynamics" by Wei et al. contains additional data and changes to the text that together with the extensive explanations in the authors' response address my earlier concerns in a satisfactory manner. Importantly, the new experiments with chemical inhibition of METTL3 are consistent with the degron mediated depletion and alleviate my concern of potential technical effects from degron methodology on the conclusion. The study has high value in providing a correction to a previous interpretation of 6NAM as part of silencing mechanism by detailing an elaborate explanation that is consistent with the majority of the published evidence (with the possible exception of understanding the function in YTHDC1 in XCI). The study also implicates a known RNA degradation mechanism in Xist turnover that has not been known before. Limitation in pinpointing the exact molecular mechanism arise from a current lack of understanding of the elements within Xist that trigger methylation dependent degradation. In conclusion, the study merits publication and will be of interest for the field of XCI and RNA biology especially researchers studying RNA modifications.

Minor points

a) New experiments were conducted following reviewer #1 criticism deleting the repA and repE proximal 6NA methylated regions in Xist which I think are very important. One would have hoped that deletion of both major 6NAM sites would confirm the results with METTL4 (and METTL14) degradation. However, the results clearly show the opposite. This is likely that the exon 7 region has additional functions in RNA stability and/or chromatin attachment and is noteworthy. Although, the main conclusion of this paper is not strengthened, it is also not weakened by these data. A brief caution from this discrepancy maybe along the lines that different regions along the Xist RNA remain to be defined in regard to their differential effects on stability.

b) In response to my earlier comment, the authors' new results in the rebuttal on WTAP are noteworthy as WTAP has been suggested to be recruited in a repeat A dependent manner (Chu et al). Yet it seems to act - as would be expected similar to METTL3 - to destabilize Xist whereas repeat A sequences are required for SPEN recruitment, stability, and gene silencing. I wonder if these opposite effects of repeat A and its binders should be mentioned as the WTAP function for 6NAM and its perceived repeat A dependent recruitment by Xist are likely to occur to readers in the XCI field. Also it appears strange to have RBM15 but not WTAP mentioned in the introduction as accessory protein of RNA methylases considering its strong effect on 6NAM.

Version 2:

Decision Letter:

4th Aug 2025

Dear Professor Brockdorff,

We are now happy to accept your revised paper "m6A and the NEXT complex direct Xist RNA turnover and X inactivation dynamics" for publication as an Article in Nature Structural & Molecular Biology.

Your paper will be published online soon after we receive proof corrections and will appear in print in the next available issue. You can find out your date of online publication by contacting the production team shortly after sending your proof corrections.

If you have not already done so, we strongly recommend that you upload the step-by-step protocols used in this manuscript to the Protocol Exchange. Protocol Exchange is an open online resource that allows researchers to share their detailed experimental know-how. All uploaded protocols are made freely available, assigned DOIs for ease of citation and fully searchable through nature.com. Protocols can be linked to any publications in which they are used and will be linked to from

your article. You can also establish a dedicated page to collect all your lab Protocols. By uploading your Protocols to Protocol Exchange, you are enabling researchers to more readily reproduce or adapt the methodology you use, as well as increasing the visibility of your protocols and papers. Upload your Protocols at www.nature.com/protocolexchange/. Further information can be found at www.nature.com/protocolexchange/about.

Authors may need to take specific actions to achieve compliance with funder and institutional open access mandates. If your research is supported by a funder that requires immediate open access (e.g. according to [Plan S principles](https://www.springernature.com/gp/open-science/plan-s-compliance) or the [NIH public access policy](https://www.springernature.com/gp/open-science/us-federal-agency-compliance)) then you should select the gold OA route, and we will direct you to the compliant route where possible. Because authors warrant under our subscription licensing terms that they haven't committed to licensing any version of their article under a licence inconsistent with the terms of our agreement – including the applicable embargo period – publication under the subscription model isn't suitable for authors whose funders require no embargo.

Sincerely,

Dimitris Typas
Senior Editor
Nature Structural & Molecular Biology
ORCID: 0000-0002-8737-1319

We thank the reviewers for their positive comments and constructive suggestions. We provide a detailed response below.

Reviewers' Comments:

Reviewer #1 (Remarks to the Author):

Summary

In this manuscript, the authors use an acute depletion method to eliminate the m6A methyltransferase and conduct a series of experiments to characterize the behavior of the Xist long noncoding RNA involved in mammalian dosage compensation. They find that acute depletion m6A increases the overall levels of Xist, promoting better X-linked gene repression. Depletion of the primary nuclear m6A reader, YTHDC1, did not phenocopy depletion of m6A. Increased Xist stability was potentially due to the disruption of the NEXT/exosome mechanism of RNA degradation.

The study uses a powerful acute depletion approach to better assess the direct effects of eliminating m6A on Xist, though this still causes all m6A to be depleted, across the transcriptome. Any potential player in this pathway that is directly and acutely influenced by loss of m6A could be involved in this mechanism. To address this major limitation of the study, the authors would benefit by more Xist-specific experiments to disrupt m6A, as well as taking advantage of previous studies from the PI that have already genetically modified regions of m6A methylation on Xist to put the current work in better context. The study is also severely limited by a lack of statistical analysis throughout the manuscript, with few exceptions. Finally, although certain pathways are ruled in and ruled out as candidates to regulate the mechanism of Xist stability, the study does not achieve mechanistic insight into what may connect m6A and NEXT, if they indeed are connected.

The following are major points that would strengthen the manuscript, if addressed:

Major Points

1. The degron approach is powerful. However, it would benefit from a control that, when acutely depleted, actually causes a significant disruption of X-linked gene repression. SPEN may be the most-likely candidate to compare. As it stands, the slight effect of YTHDC1 depletion has no appropriate point of comparison to assess if it is or is not as important as other Xist-related repression machinery.

We appreciate the reviewer's acknowledgment of the degron approach as a powerful tool to investigate the roles of Xist-interacting factors. Indeed, Dossin et al. (PMID: 32025035) employed a SPEN AID degron to elucidate a direct role for SPEN in Xist-mediated chromosomal inactivation, demonstrating a strong silencing deficiency following acute depletion. Similarly, we utilised a dTAG degron to acutely deplete PCGF3/5 (Bowness et al., 2022 PMID: 35584662) and observed a significant silencing deficiency of X-linked genes, although less pronounced than that observed for SPEN.

In order to provide a comparator for the current study we have plotted our silencing data for PCGF3/5 degron (generated in the same cell line and identical experimental setup), together with YTHDC1 degrons, see revised Figure 4d. The original plot showing individual YTHDC1 degron

clones is now shown as Extended Data Figure 9c,d. The comparison reveals that the silencing deficiency caused by YTHDC1 degron is less than that caused by PCGF3/5 degron. (Note: YTHDC1 depletion results in a small but reproducible reduction in Xist levels which does not occur following PCGF3/5 depletion).

2. It would be very helpful to show data that demonstrate the peaks of m⁶A in Xist and how they are affected by both METTL3 degrons in the context of multiple important experiments used in this study. Some of this is already published in Ref. 21. It would strengthen the manuscript to reproduce those and add to them here.

We thank the reviewer's suggestion to strengthen our study by including data on m⁶A peaks in Xist RNA and the effect of METTL3 depletion. In response, we have conducted MeRIP-seq experiments using one of the METTL3 degron lines (C3 clone) with and without dTAG-13 treatment. To comprehensively assess various conditions, we acutely depleted METTL3 for 24 hours (dTAG24h) followed by 24 hours of *Xist* induction (dTAGDox24h), and plotted this data together with our previously published analysis using dTAG2h_dTAGDox24h setup (Wei et al, 2021 PMID: 34131006). Our analysis demonstrates that 24 hours METTL3 depletion results in near complete loss of m⁶A peaks downstream of the A-repeat and E-repeat regions, consistent with what we observed previously using 2 hours dTAG treatment. *Xist* upregulation and accelerated silencing are observed as expected. This data confirms that the m⁶A peaks on Xist RNA are METTL3-dependent, strengthening the conclusions of our study. The corresponding data are described in results (lines 69-75) and presented in Extended Data Figure 1.

3. Statistical analyses are lacking in all RNA-Seq experiments beginning with Figure 1D, which limits the impact of the study.

We've added the p-value calculated from paired t-test for paired ChrRNA-seq samples. We chose paired t-test, because (1) the allelic ratio at two conditions for same gene are paired, and (2) the differences between paired values are approximately normally distributed. Please note that the number of genes analysed for each experiment affects the p-values.

4. Xist has two main regions that have been identified as m⁶A methylated on multiple adenosines, one near RepA and one Near Rep E. The Brockdorff group has previously made precise mutations in the 5' region in RepA, with little apparent effect on stability, in Ref. 18. It is surprising this was not described more clearly in the current manuscript, or used in any experiments to compare to the more extreme approach of full acute m⁶A depletion. This highlights another major limitation of this study in not being able to piece apart the detailed mechanism of which m⁶A region(s) on Xist may contribute to the stability mechanism described here. The authors should seriously consider some method of targeted disruption of the 3' m⁶A region as well. If not by genetics, then potentially by RNA-targeting Cas protein fused to FTO/ALKBH5. This would make a major contribution to strengthening the manuscript's impact, by identifying which part of Xist and which m⁶A sites may be important.

As noted by the reviewer we previously deleted the m⁶A-modified region in exon 1, downstream of Xist A-repeat (Nesterova *et al*, 2019 PMID:31311937; Coker *et al*, 2020 PMID:32258426). There

was no associated acceleration of silencing (if anything a minor reduction in silencing efficiency), or increase in Xist RNA levels. The reviewer's suggestion to investigate the second major Xist m⁶A regions in exon 7 is a good one. We opted to generate a deletion that precisely removes the major exon7 m⁶A region (153 bp), both alone and in combination with the exon 1 m⁶A region deletion. We then analysed Xi gene silencing and Xist expression levels relative to the parental cell lines (Reviewer Figure 1). As per our previous report, deletion of the m⁶A region in Xist exon 1 causes minimal (little or no) effects with respect to Xist level and XCI dynamics (Nesterova et al, 2019 PMID: 31311937). However, deletion of the m⁶A region in exon 7 led to impaired silencing and a significant reduction in Xist RNA levels. We presume this is due to transcript instability (the deletion is far from the Xist promoter region). In the double-deletion line we also observe the same phenotype as seen with exon 7 m⁶A region deletion, i.e. no additive effects.

There are several possible interpretations of these findings. The exon 7 m⁶A region may overlap with an important RNA stability element, activity of which is inhibited by m⁶A deposition. Alternatively, the deletion may impair the function of a nearby functional element, for example Xist E-repeat known to influence Xist RNA localisation, or may change the Xist splicing pattern. Finally, the deletion, by juxtaposing normally separated sequences, may change higher-order folding of Xist RNA in such a way as to impair function/stability. As we cannot discriminate between these different interpretations we have opted to show the results as a reviewer figure. Of relevance here is that other new data in response to comments by reviewers 2 and 3 fully supports our central hypothesis that METTL3-mediated m⁶A on Xist RNA functions to promote RNA turnover (see below).

Reviewer Figure 1: Analysis of Xist RNA levels and XCI dynamics following deletion of m⁶A regions of Xist exon1 and exon7.

a,b, UCSC genome browser view of m⁶A regions downstream of Xist A-repeat (**a**) and E-repeat (**b**), alongside the Xist repeat annotation and m⁶A motifs (DRACH). Xist transcription direction is indicated by the arrowhead.

c, Boxplots illustrating changes in allelic expression ratios of X-linked genes between Dox 24 h and NoDox conditions. Red lines indicate allelic ratios of 0 (no silencing) and -0.2 (representing the typical silencing level in WT after 24 h of Dox induction). Blue boxes correspond to exon7 m⁶A region deletions, yellow boxes to exon1 m⁶A region deletion, and the purple box to combined exon1 and exon7 m⁶A region deletions. Each biological replicate is represented by a separate box.

d, Barplot showing Xist RNA expression level at the indicated condition.

e, UCSC genome browser view of Xist expression profile. A summary of deletions is shown to the right. Xist RNA gene structure and transcription direction are shown at the bottom.

5. Fig. 2G: by eye, it appears there are equal, if not more, individual objects/centroids in WT vs. Degron in the Expansion images. It would be helpful to clarify why this is the case. Is this due to thresholding? The volume differences in the representative image also appear to not follow the trend from the cumulative plot. Degron does not look like a smaller volume.

Thanks for the careful evaluation of Fig. 2g (updated as Fig. 2f). We acknowledge that the chosen images may not fully reflect the trend observed in the cumulative plot. This discrepancy is not due to thresholding but rather reflects the inherent variability within the dataset, as shown in the figure, where $n > 123$ cells were analysed. We have updated these representative images accordingly. In rechecking these datasets we noticed that we inadvertently used a dataset in which signal intensity was suboptimal. Accordingly we used a second dataset in which both pulses were optimal to replot as shown in Figure 2e. The suboptimal dataset was not used in any other analyses shown in this study. The re-analysis revealed that there is a significant increase in cloud volume both at expansion and steady state phases. We have modified the text accordingly (lines 179-183).

6. Fig. 3C: a plot of total Xist would be informative, at least in the supplemental, to determine if, in this window, the absolute amount of Xist is compromised, or if the exchange of old for new is the primary point of disruption.

We agree with the referee that Fig. 3c does not display total Xist RNA. To address this using our turnover dataset, we have plotted total Xist centroid levels at the start and end of each time course. Total Xist levels follow the same trend in both WT and METTL3 degron conditions during both expansion and steady state phases. Throughout the time course, there are more Xist centroids as a consequence of METTL3 depletion stabilising Xist in both phases (similar to that shown in 2d). (In expansion, total Xist centroid count continues to increase, whereas as steady state progresses there is a decrease in total Xist centroids, as published previously Rodermund et al. 2021 PMID:34112668). This suggests that whilst new Xist molecules are still made and can be exchanged for old molecules (see also Fig. 3b), increased stability of Xist molecules is the predominant cause of increased Xist centroid number in the METTL3 degron line. This analysis is now included as Extended Data Fig. 6.

7. As a point of discussion, is it helpful to address the “coupling” phenomenon that the Brockdorff group described in Ref. 25.

Coupling of pre-synthesised and newly synthesised Xist molecules is indeed observed after METTL3 depletion and this is evident in Figure 3b, for example at the 100' timepoint. We weren't able to identify a point in the discussion where it would be logical to introduce this observation without distracting from the key points being made.

Minor Points

- A simplification of the clone names, perhaps simple “Clone #1 and Clone #2” would help reduce the need to keep track of alphanumeric clone names.

Some of the clones used in this study were first reported in our prior paper describing METTL3 degrons and for the sake of consistency we have retained our labelling system. We have added a

summary table (Extended Data Figure 12) to aid readers in tracking the modifications associated with the different clone names.

- Figure 1d: is there an appropriate control that can be shown with dTAG-13 ligand in the C7H8 parental line, +/- dox? This would better match the rest of the figure. In Figures 1d and 2a the parental line has replicates with no dTAG-13 instead, which does not match the rest of the experiments well.

Thanks for pointing this out. We have performed the experiment and updated the corresponding figures. See updated Fig. 1d and 2a.

- A better introduction to the Expansion and Steady-State windows would help the reader appreciate these two distinct phases.

Thanks for the suggestions. We've added the following sentences, lines 175-178;

'In prior work using this system we reported that Xist RNA accumulates to maximal levels of around 50-100 molecules/cell over a period of 1.5 and 5 hours, referred to as expansion phase. 24 hours was selected as a timepoint where Xist RNA levels have attained a steady state'

- Line 177: the "*" likely refers to just one of the two p-values. Either generalize or add specific values for each plot.

Done.

- Figure 2d can go in supplement.

We moved Figure 2d to Extended Data Fig. 2a.

- Line 285-287: the PC analysis could be better described here.

We have amended our sentence (lines 307-309) accordingly.

"Principal component analysis indicates that accelerated silencing affects X-linked genes equivalently across the X chromosome, which contrasts with the silencing deficiency observed upon knockout of the key silencing factor SPEN or deletion of B/C repeats in Xist (Extended Data Fig. 11d)."

- Line 353-355: is this model described in reverse? Wouldn't m6A promote the association of RBPs that would promote NEXT/exosome activity?

Thanks for the careful reading and pointing this out. We have corrected the figure.

- When discussing m6A effects on lncRNA stability, it may be helpful to reference studies that found m6A sites can promote nuclear lncRNA stability, as a point of comparison (PMID 34048709; PMID 36441764).

Thanks for the suggestion. Both studies (PMID: 34048709 and PMID: 36441764) reported that m⁶A promoting nuclear lncRNA stability are YTHDC1 dependent. Note that to a certain extent our findings that YTHDC1 may promote Xist stability agrees with this notion. We added the following sentence into the Discussion (lines 380-382);

"Together with previous reports showing that m⁶A can promote nuclear RNA stability via YTHDC1^{42,43}, our findings underscore the complexity and context-dependent nature of m⁶A function in regulating nuclear RNA stability. Additional studies are required to fully elucidate the factors and pathways governing m⁶A-mediated nuclear RNA stability."

Reviewer #2 (Remarks to the Author):

The authors of this manuscript report a careful analysis of the roles of m6A on Xist function. This manuscript shows a destabilizing effect of m6A on Xist RNA. In contrast to a previous report acute METTL3 depletion (m6A depletion) led to increased Xist level and X inactivation activity independent of YTHDC1. The authors also showed that the NEXT complex (ZCCHC8) is involved in Xist turnover. The results are convincing and should be published. I do have several comments:

1. There is a very good METTL3 inhibitor developed by Tony Kouzarides and colleagues. I **strongly** suggest applying this inhibitor for acute inhibition of METTL3. This could resolve the concern of WT versus mutant rescue and convincingly show the activity dependency of Xist RNA turnover regulated through m6A.

Thanks for the valuable suggestion. To address this point, we performed experiments using STM2457 to inhibit METTL3's catalytic activity. We applied three different concentrations (10 μ M, 20 μ M, and 30 μ M) and observed consistent results with the dTAG degron approach. Inhibition of METTL3 activity led to increased Xist levels and accelerated XCI dynamics, with higher concentrations producing stronger effects. Additionally, inhibition at 20 μ M and 30 μ M had a more pronounced and reproducible impact on *Tor1aip2* intron 3 splicing compared to 10 μ M, further supporting the dose-dependent role of METTL3 catalytic activity (note that the phenotype observed upon 30 μ M METTL3i treatment is weaker than that of METTL3 degron). These findings reinforce our conclusion that Xist RNA turnover is regulated by METTL3 enzymatic function. The experiment is described in results, lines 150-158 and illustrated in Extended Data Figure 4.

2. This one could be optional. There is a quite selective inhibitor of YTHDC1.

See: <https://www.researchsquare.com/article/rs-2644364/v1>. This molecule works quite well in our hands in cell-based studies. It would be nice to synthesize this molecule and test acute inhibition of YTHDC1 in the current system.

Thank you for the suggestion. We performed YTHDC1 inhibition using compound 11 (Cat# Z287226108, Enamine Ltd) at 50 μ M, as referenced in PMID: 34375583, with three independent biological replicates. While we observed a consistent minor acceleration of XCI dynamics, Xist level changes were inconsistent across replicates. In those replicates (rep1 and rep3) where Xist was indeed upregulated, the effect (1.11–1.13 fold increase) was remarkably weaker than METTL3 inhibition at 20 μ M (1.297 fold) and 30 μ M (1.612 fold). Additionally, *Tor1aip2* intron3 splicing results varied among replicates, suggesting that YTHDC1 inhibition does not fully recapitulate METTL3 inhibition. Notably, while inhibition of YTHDC1 had inconsistent effects on this splicing event, conditional YTHDC1 knockout closely mirrored METTL3 knockout or inhibition across multiple datasets (PMID: 34131006). These findings support that YTHDC1 plays a minor role in Xist RNA turnover and is not the sole m⁶A reader regulating this process. See Reviewer Figure 2.

a**b****d***Tor1aip2* splicing

	n_intron2	n_intron3
Rep1_YTHDC1i_50uM	2	2
Rep2_YTHDC1i_50uM	12	2
Rep3_YTHDC1i_50uM	0	1

Reviewer Figure 2: Xist levels and XCI dynamics upon inhibition of YTHDC1.**a**, Schematic outlines the experimental design.**b**, Boxplot depicting the allelic ratio of X-linked genes for three independent biological replicates. The red line marks an allelic ratio of 0.5. P-values are calculated using a paired *t*-test. Sample conditions are indicated below.**c**, Barplot showing *Xist* levels, with arrowhead indicating the up-or-downregulation and fold-change labelled.**d**, Table summarising the number of reads spanning intron2 and intron3 of *Tor1aip2* gene.

3. It would be nice if the authors can KD a few selected m⁶A binding proteins to see if they are involved in Xist degradation. hnRCPC, hnRNPA2BA, FXR are obvious candidates. I think it will be nice to test a few known candidates. If they do not work at least the authors can exclude them.

Many RNA-binding proteins, including several known m⁶A direct and indirect readers (such as YTHDC1, IGF2BP1, IGF2BP3, HNRNPC, and HNRNPA2B1, though not FXR), have indeed been identified as Xist interactors through Mass Spectrometry analysis (ChIRP-MS) (Chu et al., Cell 2015 PMID:25843628). While knocking down selected m⁶A-binding proteins to assess their role in Xist RNA stability is an interesting suggestion, it presents certain challenges given the potential cooperative and/or combinatorial effects of multiple readers on Xist m⁶A, as well as the inherent limitations of knockdown approaches. We believe this strategy may not yield conclusive results. Nonetheless, we acknowledge the importance of the question. Clearly future studies will be necessary to systematically dissect how individual m⁶A readers contribute to Xist RNA degradation.

Reviewer #3 (Remarks to the Author):

The manuscript entitled "N6-methyladenosine and the NEXT complex direct Xist RNA turnover and X inactivation dynamics" by Wei et al. investigates the role of RNA methylation on Xist function and the initiation of X chromosome inactivation in a mouse embryonic stem cell model. The authors use degron mediated depletion of the catalytic subunit of the N6-adenine methyltransferase complex to study the requirement for the establishment of X-linked gene repression, which in the study is mediated by Doxycycline regulated Xist expression. A main finding of the study is that Xist initiates efficient gene repression after depletion, which corrects earlier conclusions on a requirement of N6-A RNA methylation for gene silencing. The primary effect of RNA methylation appears destabilization of Xist RNA and after depletion Xist levels are increased at early timepoints after induction which leads to slightly increased or faster gene silencing. The authors go on to investigate Xist turnover by two independent methods RNA split using 3D-SIM superresolution microscopy and SLAMseq. Both methods show stabilization of Xist after Mettl3 depletion. Mechanistically Mettl3 appears to trigger degradation via a pathway involving ZCCHC8 but requires neither ZFC3H1 nor YTHDC1. The study provides convincing genetic evidence using degron-tagged alleles of the respective genes. However, the link to RNA methylation remains to be fully understood in future studies.

The paper stands out for the large amount of sophisticated evidence and clear conclusions. It is important for correcting the role of RNA methylation for Xist function. Although, the authors have already voiced concerns over a requirement in gene silencing this study appears to fully solve the issue and is important for the field. A few point that need clarification should be considered by the authors to further improve the presentation of the study.

Specific points

1. The authors find that Xist becomes more stable after depletion of Mettl3. If Mettl3 indeed were a major pathway for Xist degradation one could expect that Xist accumulates in the nucleus and potentially localizes to other chromosomes. The authors argue that this is not observed because of a negative effect on Xist transcription. I did not fully understand how this effect should operate. Firstly, Xist expression is said to be controlled by a "TetON" promoter and this raises the suspicion that it might not be fully relevant for regulation of the Xist promoter. Secondly, references to a negative effect on Xist transcription are to the authors own previous work but do not identify a plausible mechanism. It would be important to clarify this aspect in the manuscript.

This remark is in reference to comments in the discussion, lines 354-356. In ongoing work, we have clear evidence for a role for Xist RNA-dependent SETDB1/HUSH recruitment acting to dampen Xist transcription, both for the native Xist promoter and heterologous TetON system. Acute depletion of SETDB1/HUSH results in a dramatic increase in Xist RNA levels and the rate of Xi silencing. Thus, we consider that this pathway is at least one component of the feedback mechanism that links Xist RNA stability and transcription. We are currently writing a manuscript describing these findings. However, as this work is currently unpublished we have simplified this part of the discussion, substituting in the sentence;

'The basis for feedback control between Xist transcription and turnover remains unknown and is an important topic for future studies'.

2. Moindrot et al. (REF 10) and Chu et al. (REF 8) also identified WTAP as an Xist associated protein, which links to RNA methylation. Chu et al. show that WTAP is recruited to Xist dependent on gene repression. It would be interesting to consider WTAP either as a potential link between RNA methylation and gene silencing or explain why this has not been considered in the study.

Structural and biochemical studies have established that within the m⁶A writer complex, METTL3 is the only catalytic subunit, while METTL14 has a degenerate active site and primarily supports complex integrity and RNA substrate recognition. WTAP is a well-characterized and essential accessory subunit of the m⁶A writer complex, directly interacting with METTL3. Multiple studies across species have demonstrated that WTAP knockout leads to a global reduction in m⁶A levels. Although WTAP has been identified as an Xist-associated protein, it is important to note that WTAP does not possess any known m⁶A reader domains, either canonical (e.g., YTH domain) or noncanonical. For these reasons we didn't initially consider analysis of WTAP function in this context. However, prompted by the reviewers question we attempted to insert an FKBP12^{F36V} tag in-frame at the WTAP N-terminus (Reviewer Figure 3). PCR verification confirmed successful knock-in for two independent clones, but western blot analysis failed to detect the fusion protein (despite use of a highly specific and reliable anti-WTAP antibody). These observations imply that the N-terminal FKBP12^{F36V} tag knockin may interfere with WTAP protein stability, which leads either to full degradation or to a strongly reduced level even in the absence of dTAG-13 reagent, or disrupts the first intron splicing of *Wtap*, which generates an unstable transcript. Consistent with this, *Top1aip2* splicing analysis indicated globally low m⁶A levels in both clones. Moreover, upon inducing Xist expression for 24 hours in these clones, we observed increased Xist RNA levels and accelerated XCI dynamics compared to the parental cell line, phenocopying METTL3 and ZCCHC8 depletion. These findings support a role for WTAP in Xist regulation similar to METTL3, consistent with it being an essential subunit of the m⁶A writer complex assembly. As regulated acute depletion wasn't possible using these cell lines, we feel it is more appropriate to show this data as a reviewer figure only.

Reviewer Figure 3: Xist levels and XCI dynamics in FKBP12^{F36V}-WTAP cells.

a, Strategy showing in-frame insertion of FKBP12^{F36V} into WTAP.

b, Validation of FKBP12^{F36V} insertion in the selected clones using PCR.

c,d, Schematic and barplot showing the ratio of the number of reads spanning intron2 and intron3 of *Tor1aip2* gene.

e, Boxplot depicting the allelic ratio of X-linked genes for parental line and FKBP12^{F36V}-WTAP. The red line marks an allelic ratio of 0.5. P-values are calculated using a paired *t*-test. Sample conditions are indicated below.

f, Barplot showing *Xist* levels for selected clones.

3. The authors show that METTL14 protein is strongly reduced after degron-mediated depletion of METTL3. This brings up the concern that the degron might affect not only the target but also associated proteins, which in the context of chromatin could potentially also include components of the transcription machinery. As the present study seems to contradict earlier conclusions of a requirement of METTL3 for gene silencing, it would be important to consider such effects. Would chemical inhibition of the METTL3/14 complex also lead to Xist stabilization and more rapid or effective silencing?

Our observation that acute depletion of METTL3 causes massive reduction of METTL14 protein level agrees with a recent report that METTL3 protects METTL14 degradation (Zeng et al, EMBO Report 2023, PMID: 36597993). This actually aligns with the known heterodimeric nature of METTL3 and METTL14.

To reinforce our conclusion that Xist RNA turnover is regulated by METTL3 enzymatic function, we have further performed experiments using STM2457 to inhibit METTL3 catalytic activity. We applied three different concentrations (10 μ M, 20 μ M, and 30 μ M) and observed consistent results with the dTAG degron approach. Inhibition of METTL3 activity led to increased Xist levels and accelerated XCI dynamics, with higher concentrations producing stronger effects. Additionally, inhibition at 20 μ M and 30 μ M had a more pronounced impact on *Tor1aip2* intron 3 splicing compared to 10 μ M, further supporting the dose-dependent role of METTL3's catalytic activity. See also the response to Reviewer#2 comments, as well as Extended Data Figure 4.

Minor points

a) Can cell cycle effects be ruled out to lead to an increase of Xist RNA after acute depletion of Mettl3? As Xist displacement from chromatin in M-phase contributes to turnover a statement could be added if cell cycle differences might have been observed.

We believe that cell cycle effects are unlikely to contribute significantly to Xist RNA turnover within the 26-hour timeframe of our experiments. Notably, we observed that Xist exhibits a significantly longer half-life compared to wild-type during the expansion phase, only 1.5 hours after induction (Figure 3). Given that this timescale is too short for substantial cell cycle changes to occur, the observed effects are likely independent of cell cycle dynamics.

b) I found it puzzling that RNA methylation but not YTHDC1 had an effect on Xist stabilization. It would be interesting to have the authors' opinion on what YTHDC1 function might be or indeed if it binds Xist.

We were also puzzled by this finding given data suggesting a role for YTHDC1 in promoting turnover of m⁶A-modified carRNAs and PROMPTs and conversely, in protecting m⁶A-containing RNAs from degradation through nuclear condensate formation (a discussion of how YTHDC1 may differentially regulate m⁶A-containing transcripts to either enhance stability or promote degradation, depending on the cellular context is provided in PMID: 38629637). We observed clear effects of YTHDC1 depletion on splicing regulation, for example *Tor1aip2* intron3 and *Spn* intron2, that phenocopy those seen for METTL3 depletion (Wei et al, 2021 PMID: 34131006; Supplemental Figure 12, 14, and 15). Thus, to directly answer the reviewer's question, we believe that YTHDC1 is important to mediate m⁶A-dependent alternative splicing, and in some cases in regulating m⁶A-dependent RNA stability, albeit not in the case of *Xist* (and *Kcnq1ot1*). It is clear that YTHDC1 does bind Xist RNA, supported by several independent mass spectrometry experiments. Whether or not it has a role in regulating Xist RNA function remains unclear.

c) line 285: after "5d" the closing parenthesis seems missing

Done.

We thank all the reviewers for their constructive comments that help improve this study.

Reviewer #1 (Remarks to the Author):

This revised manuscript has met all of the suggestions that this reviewer deemed important for publication. Although this is at the author's discretion, this reviewer encourages the authors to at the very least include a mention of the exon 7 m6A site deletion result that was described only in the rebuttal. Even with the somewhat complex possibilities for interpretation, the result generally fits with the model and suggests a possible RNA stability sequence that is negatively regulated by m6A, which the authors note. This information is valuable to the field.

We thank Reviewer #1 for the positive feedback on the experiments. In response to the shared interest from both Reviewer #1 and #3, we have now discussed this point in the revised manuscript (See Discussion) and included the corresponding results as Supplementary Fig. 2.

Reviewer #2 (Remarks to the Author):

The authors have addressed my comments.

Reviewer #3 (Remarks to the Author):

The revised version of the manuscript entitled "N6-methyladenosine and the NEXT complex direct Xist RNA turnover and X inactivation dynamics" by Wei et al. contains additional data and changes to the text that together with the extensive explanations in the authors' response address my earlier concerns in a satisfactory manner. Importantly, the new experiments with chemical inhibition of METTL3 are consistent with the degron mediated depletion and alleviate my concern of potential technical effects from degron methodology on the conclusion. The study has high value in providing a correction to a previous interpretation of 6NAM as part of silencing mechanism by detailing an elaborate explanation that is consistent with the majority of the published evidence (with the possible exception of understanding the function in YTHDC1 in XCI). The study also implicates a known RNA degradation mechanism in Xist turnover that has not been known before. Limitation in pinpointing the exact molecular mechanism arise from a current lack of understanding of the elements within Xist that trigger methylation dependent degradation. In conclusion, the study merits publication and will be of interest for the field of XCI and RNA biology especially researchers studying RNA modifications.

Minor points

a) New experiments were conducted following reviewer #1 criticism deleting the repA and repE proximal 6NA methylated regions in Xist which I think are very important. One would have hoped that deletion of both major 6NAM sites would confirm the results with METTL4 (and METTL14) degradation. However, the results clearly show the opposite. This is likely that the exon 7 region has additional functions in RNA stability and/or chromatin attachment and is noteworthy. Although, the main conclusion of this paper is not strengthened, it is also not weakened by these data. A brief caution from this discrepancy maybe along the lines that different regions along the Xist RNA remain to be defined in regard to their differential effects on stability.

We thank Reviewer #3 for the positive feedback on the experiments. In response to the shared interest from both Reviewer #1 and #3, we have now discussed this point in the revised manuscript (See Discussion) and included the corresponding results as Supplementary Fig. 2.

b) In response to my earlier comment, the authors' new results in the rebuttal on WTAP are noteworthy as WTAP has been suggested to be recruited in a repeat A dependent manner (Chu et

a). Yet it seems to act - as would be expected similar to METTL3 - to destabilize Xist whereas repeat A sequences are required for SPEN recruitment, stability, and gene silencing. I wonder if these opposite effects of repeat A and its binders should be mentioned as the WTAP function for 6NAM and its perceived repeat A dependent recruitment by Xist are likely to occur to readers in the XCI field. Also it appears strange to have RBM15 but not WTAP mentioned in the introduction as accessory protein of RNA methylases considering its strong effect on 6NAM.

Thanks for the valuable suggestion.

(1) We have added the following sentence into the discussion.

“Intriguingly, the Xist A-repeat region is bound by RBPs such as SPEN, which promotes Xist RNA stability, and RBM15/WTAP, which contribute to its destabilisation through m⁶A modification. The opposing effects of these pathways on Xist RNA stability highlight a regulatory balance that warrants further investigation.”

(2) We have mentioned WTAP in the introduction and thus modified the following sentence.

“Other subunits of the m⁶A writer complex, particularly WTAP, were also identified in the Xist proteomic and functional screening experiments. Notably, the recruitment of both RBM15 and WTAP to Xist depends on the Xist A-repeat region.”